# Thermal and Mechanical Properties of Nano-Carbon-Reinforced Polymeric Nanocomposites: A Review

**Zeeshan Latif [1], Mumtaz Ali [1], Eui-Jong Lee [2], Zakariya Zubair [1,*] and Kang Hoon Lee [3,*]**

[1] School of Engineering and Technology, National Textile University, Faisalabad 38000, Pakistan; zeeshanlatif203@yahoo.com (Z.L.); mumtaz.ali@ntu.edu.pk (M.A.)

[2] Department of Environmental Engineering, Daegu University, 201 Daegudae-ro, Jillyang, Gyeongsan-si 38453, Republic of Korea; lujong@daegu.ac.kr

[3] Department of Energy and Environmental Engineering, The Catholic University of Korea, 43 Jibong-ro, Bucheon-si 14662, Republic of Korea

* Correspondence: zzubair@ntu.edu.pk (Z.Z.); diasyong@catholic.ac.kr (K.H.L.)

**Abstract:** Carbon nanomaterials are an emerging class of nano-reinforcements to substitute for metal-based nanomaterials in polymer matrices. These metal-free nano-reinforcement materials exhibit a high surface area, thermal stability, and a sustainable nature. Compared to conventional reinforcements, nano-carbon-reinforced polymer composites provide enhanced mechanical and thermal properties. While previous reviews summarized the functionality of nanocomposites, here, we focus on the thermomechanical properties of nano-carbon-reinforced nanocomposites. The role of carbon nanomaterials, including graphene, MXenes, carbon nanotubes, carbon black, carbon quantum dots, fullerene, and metal–organic frameworks, in polymer matrices for the enhancement of thermal and mechanical properties are discussed. Different from metal-based nanomaterials, carbon nanomaterials offer high specific strength, abundance, and sustainability, which are of considerable importance for commercial-scale applications.

**Keywords:** carbon nano-reinforcements; MXene; mechanical properties; thermal resistance; polymers

## 1. Introduction

Polymer composites reinforced with carbon nanomaterials are receiving interest for their sustainable and abundant nature, along with their provision of the intimate interface required for outstanding properties. Consequently, to maximize their performance, it is crucial to grasp the reinforcing mechanisms [1–5]. To overcome certain limitations of polymers, different carbon-based nanofillers are frequently used as reinforcements for polymers [6–12]. An innovative replacement for traditional polymer composites has been made possible by using nanoscale fillers to improve the thermal, mechanical, and physical characteristics of polymers. Nanoscale fillers range fundamentally from isotropic to highly anisotropic sheet-like or needle-like morphologies, with a minimum of a single distinctive length scale in the order of nanometers. Innovative combinations of nanoscale materials and polymer matrices to produce polymer nanocomposites with intriguing characteristics are made possible via interactions at a molecular level [5].

Based on the dimensions of dispersed carbon nanomaterials, there are three primary types of polymer nanocomposites. The first type of carbon-based nanofillers is two-dimensional (2D) reinforced nanomaterials, including reduced graphene oxide (rGO) and MXenes [4]. Graphene and MXene sheets are a few nanometers thick, whereas carbon-based nanofillers of other dimensions are much thicker. The second class of carbon-based nanofillers is one-dimensional (1D) reinforced carbon nanotubes (CNTs), including different types, e.g., single-walled carbon nanotubes (SWCNTs) and multi-walled carbon nanotubes (MWCNTs). These 1D nanofillers are employed as nano-reinforcing materials in

polymer matrices to achieve better characteristics of nanocomposites [13]. The third group of carbon-based nanomaterials is zero-dimensional (0D), including carbon quantum dots (CQDs), fullerene, and carbon black, which offer different particle shapes. Due to their small size and tunable structural properties, these nano-reinforcements are emerging as a means of obtaining nanocomposites with exceptional properties [14,15]. Another class of carbon-based nanomaterials is metal–organic frameworks (MOFs), which have no specific dimensions. The structures of all carbon-based nanofillers, including graphene, MXenes, CNTs, CB, CQDs, fullerene, and MOFs, are shown in Figure 1.

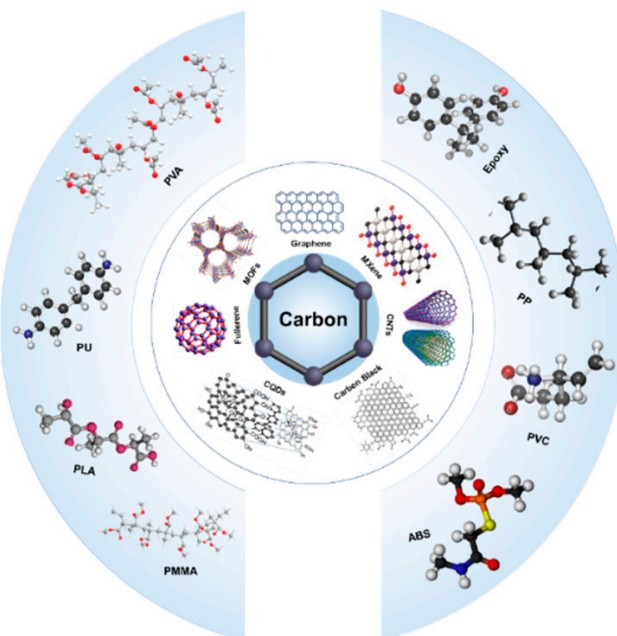

**Figure 1.** Different structures of carbon derivatives.

Recently, studies on the fabrication of nanocomposites using derivatives of nano-carbon in polymers demonstrated their superior mechanical, thermal, and functional characteristics. The characteristics of nano-carbon reinforcements that enhance nanocomposites include (i) a high aspect ratio, as well as strong interfacial interactions with polymer chains, (ii) homogeneous dispersions of nanofillers in the polymer, (iii) higher bonding strength between nanofillers and the polymer, (iv) the crosslinking behavior of nanofillers with matrices, (v) greater mechanical interlocking of nanofillers and matrices, (vi) strong intermolecular interactions of these nanofillers with polymer matrices, suppressing the free vibrations and hence reducing heat conduction. On the basis of these characteristics, the thermal and mechanical properties of nano-carbon-reinforced polymer nanocomposites can be enhanced, as highlighted in this review. Previous reviews have focused on metal-based nanoparticles used as nano-reinforcement materials in polymer matrices, and the functional aspects of nanocomposites were studied [16–19]. On the contrary, here, we emphasize the reinforcement effect of nano-carbons, specifically the thermal and mechanical properties of nano-carbon-reinforced polymer composites.

In the emerging commercial applications of polymers, thermomechanical properties are of critical importance. The life and performance of polymer composite parts are determined based on thermomechanical properties. Therefore, it is important to understand the recent progress and mechanisms related to the thermal and mechanical properties of carbon nanofillers such as graphene-, Mxene-, CNT-, CB-, CQD-, fullerene-, and MOF-based polymer nanocomposites. The dispersion of these carbon nanoparticles and their interactions with polymer matrices that improve the thermal and mechanical performance of polymeric nanocomposites are summarized. Based on our understanding of the literature,

future directions in nano-carbon research for the enhanced performance of nanocomposites are provided.

## 2. Graphene-Based Nanocomposites

Graphene and other graphene-based substances have established the vital components of composite technology. Graphite is exfoliated and oxidized to synthesize graphene oxide (GO), one of the most structurally significant compounds. This is then reduced to obtain rGO, also employed in composites, as the removal of the oxide groups might lead to fewer structural flaws [20]. Various nanocomposites have been fabricated using rGO as nano-reinforcement to analyze their thermal and mechanical properties. For instance, the impact of filler insertion techniques on the electrical and mechanical characteristics of poly(methyl methacrylate) (PMMA) nanocomposites embedded with rGO was studied [21]. In situ polymerization of the MMA monomer with the inclusion of rGO results in the formation of PMMA beads/rGO and in-situ polymerization of MMA on the surface of rGO. The impact of adding different contents (between 0.1 and 2% *w/w*) of rGO on the thermal and mechanical characteristics of PMMA was examined. At a 1 wt.% concentration of rGO, the thermal conductivity of the PMMA/rGO nanocomposite was enhanced by 126% as compared to pure PMMA [21].

In another work, rGO/polyvinyl alcohol (PVA) nanocomposites were fabricated via coagulation of the mixture with 2-propanol [22]. The composite structure showed enhanced connections, mostly involving hydrogen bonds, formed between the polymer and the rGO sheets. The glass transition temperature ($T_g$) of the PVA/rGO nanocomposite was improved by 25% with the inclusion of 10 wt.% rGO [22].

Because of their complementary functional characteristics, materials based on micro- and nano-fibrillated cellulose (MFC/NFC) and graphene have potential in a range of fields [23]. Using rGO as reinforcement to fabricate versatile MFC nanocomposites provided a straightforward water-dispersion-based mixing process. The characteristics of the MFC composites varied according to the filler type. Young's modulus rose from 18 Gpa to 25 Gpa, while the tensile strength of MFC/GO and MFC/rGO nanocomposites was enhanced by 18% and 23% by using 0.6 wt.% reinforcement content, respectively (Figure 2a). In comparison to pure MFC, the inclusion of 5 wt.% and 2 wt.% rGO raised the thermal stability by 5% and 2%, respectively (Figure 2b) [23].

Nanocomposites of phenol formaldehyde (PF)/rGO were fabricated with different loadings of rGO to observe the thermal and mechanical properties [24]. By efficiently integrating rGO into PF resin, different processing parameters were optimized. The good dispersion of rGO and the structure of rGO and the PF/rGO nanocomposite were confirmed using transmission electron microscopy (TEM). Thermogravimetric analysis (TGA) was used to assess the impact of rGO on the thermal characteristics of the polymer nanocomposites. The thermal resistance of the PF/rGO nanocomposite was enhanced by 12% with the incorporation of 0.08 wt.% rGO as compared to pure PF, as shown in Figure 2c. With the inclusion of 0.12 wt.% rGO, the overall mechanical performance was increased by 150%, as compared to pure PF (Figure 2d) [24]. By using a simplified solution-mixing–evaporation technique, sodium carboxymethyl cellulose (NaCM)/rGO and NaCMC/GO composite films were fabricated. The strength and modulus of CMC/rGO were significantly increased by 73% and 132%, respectively, compared to pristine CMC upon the addition of 2 wt.% rGO (Figure 2e) [25].

Using the solution-casting process, polyvinylidene fluoride (PVDF) and rGO nanocomposites were formed [26]. The inclusion of rGO significantly alters the structure of PVDF; the results revealed that the ß-phase proportion was enhanced at the cost of the α-phase. According to differential scanning calorimetry (DSC) measurements, adding more rGO reduces the crystallinity of nanocomposites [26].

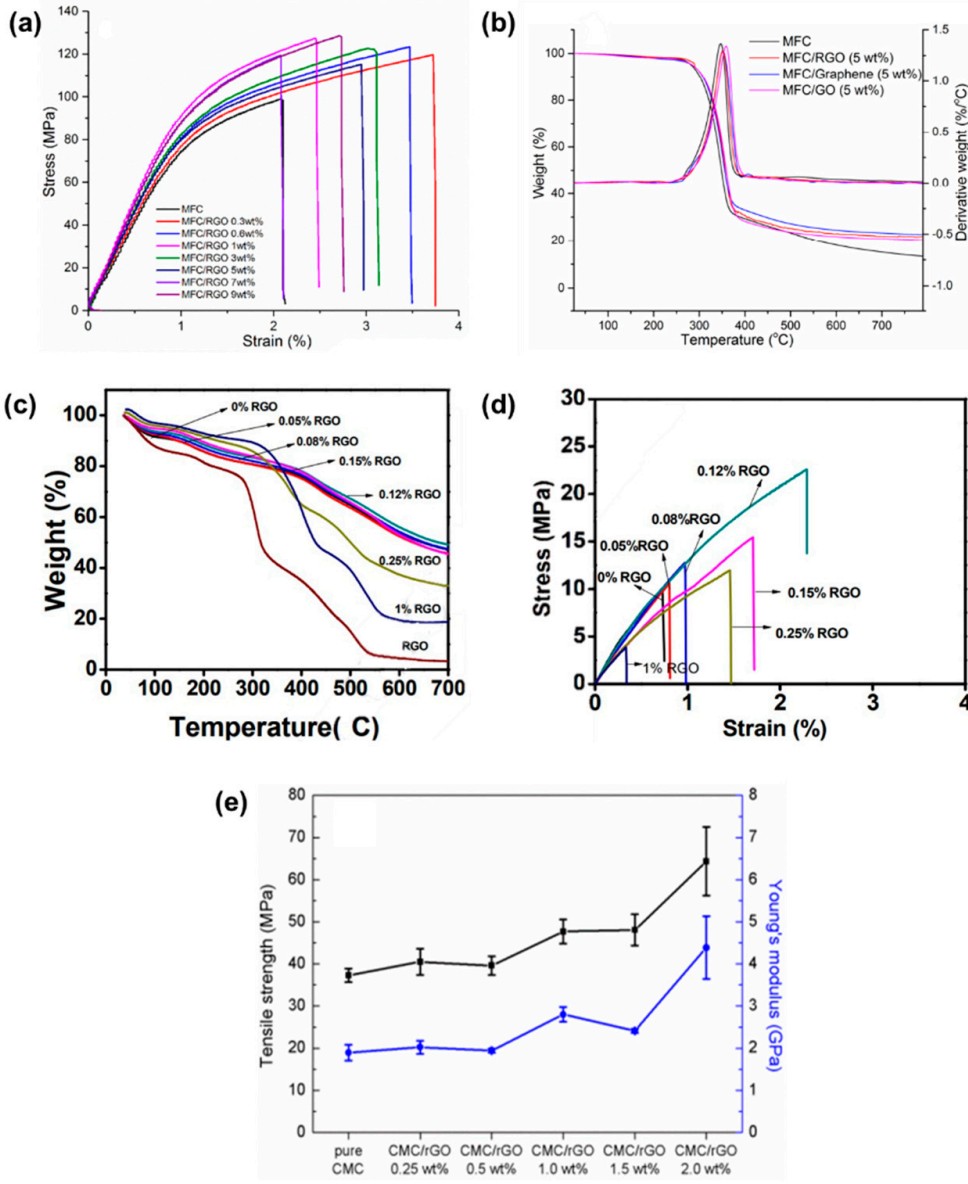

**Figure 2.** (**a**) The stress–strain curves and (**b**) thermal degradation of nanocomposites [23]. (**c**) Thermal resistance and (**d**) mechanical properties of PF/rGO nanocomposites [24] (**e**) Tensile strength and Young's modulus of CMC/rGO [25].

To address the issues of rGO dispersion and the thermal resistance of epoxy resin, additive-type fire-resistant piperazine-rGO/piperazine-based DOPO-phosphoramidite (PiP-DOPO) was studied [27]. Piperazine-functionalized rGO was subsequently integrated into PiP-DOPO via an in-situ process to create hybrids (PD-rGO). PD-rGO was then mixed with epoxy to create a nanocomposite. The composite structure increased thermal stability up to 720 °C and decreased the maximum weight loss (under $NO_2$) due to the inclusion of PD-rGO in epoxy. By adding 4 wt.% PD-rGO to epoxy, as compared to clean epoxy, the peak heat release rate and total heat release quantities were dramatically decreased by 43% and 30.2%, respectively, according to the analysis of burning behavior. The limiting oxygen index (LOI) was 28%, and the epoxy composite with 4 wt.% PD-rGO exceeded that of the UL-94 V0 grade [27]. The combination of binary arrangements in the PD-rGO hybrid, PiP-DOPO, and graphene produced a fire-resistant effect that might be related to their synergistic effects on flammability prevention in the UL-94 burning assessment and heat release rate reduction, respectively. Because of the high rigidity of graphene in the hybrids, the inclusion of PD-rGO also resulted in a slightly enhanced modulus and strength.

High-performance epoxy nanocomposites with increased fire resistance and mechanical capabilities at the same time were fabricated with PD-rGO, which integrates the exceptional mechanical performance of graphene with the positive fire-resistant impact of DOPO-based substances [27].

An analysis was conducted to check various compatibilizers that affected the characteristics of rGO-reinforced polyethylene terephthalate/polybutylene terephthalate (PET/PBT) nanocomposites [28]. Injection molding and melt-compounding techniques were used to form the specimens. At various loading levels (0.5–4%), the compatibilizers Joncryl and glycidyl isooctyl polyhedral oligomeric silsesquioxane (GPOSS) were employed. The thermal data demonstrated that the inclusion of the compatibilizer slightly changed the blend's thermal stability, whilst the inclusion of rGO had no impact. With the introduction of 2 wt.% GPOSS, the nanocomposite's $T_g$ dropped by 33%. Also, the tensile test showed a 17% enhancement in the mechanical properties of PET/PBT/rGO with the incorporation of 0.5 wt.% compatibilizers in PET/PBT [28].

A nanocomposite fabricated by grafting rGO with self-synthesized 10 wt.% poly caprolactone-diols (PCL-diols) into polyurethane (PU) was fabricated using a tailored Hummer's method [29]. The grafted PCL-diols widen the interplanar spaces of rGO and make it easier for PU chains to be introduced between nanosheets, which improves rGO nanosheet compatibility and dispersion in the PU matrix. Additionally, they reduce the crystal size and melting temperature ($T_m$) of soft portions while changing the crystalline structure of PU [30]. Functionalized graphene oxide (FGO) combines with a specific kind of functional group in PU molecules to create a 3D network of polymers, along with reinforcement [31,32]. The thermal properties and Young's modulus of the PU/FGO nanocomposite were enhanced by 63% and 58% with the addition of 1 wt.% FGO [29].

An isocyanate group/FGO was incorporated into PU prepolymers with diselenide linkages that were hydroxyl-terminated, and then in-situ polymerization was used to form a nanocomposite [33]. In the composite with 2 wt.% FGO, the mechanical properties and $T_g$ were increased by 54.5% and 8%, respectively. Due to diselenide linkages, the nanocomposite demonstrated shape memory and repetitive self-healing when exposed to NIR light. It could perform dynamic bond exchange under certain circumstances and had a comparatively low bond energy of 172 kJ/mol. After introducing a diselenide element, the nanocomposite demonstrated 90% healing efficiency, which stayed above 75% after five healing cycles. This may be attributable to the cooperative action of the photothermal reaction, shape memory effect (SME), and dynamic interchangeable diselenide attachment, which helped to close the affected fracture and accomplish healing [33].

Diethyl-toluene-diamine (DETDA) and diglycidyl ether bisphenol A (DGEBA) matrices/graphene oxide nanosheet (GON) nanocomposites were studied for their thermal stability and mechanical properties. At various ratios of GON and starting pressures (IPs), physical variables such as the stress–strain curve, order parameter, and atomic length extension were studied. The slope of the stress–strain curve increases as the atomic ratio of GONs increases. Thus, atomic structures have a higher Young's modulus. The order parameter is increased, and interatomic interactions are optimized by increasing the GON atomic ratio by up to 5%. Additionally, the atomic length extension of GONs is reduced from 11 to 8 by raising the atomic ratio from 1 to 5%. The stress–strain curve's slope rises from 2.91 to 3.55 GPa as the IP rises. These investigations demonstrate that the order parameter is reduced from 0.48 to 0.4 as the IP grows from 1 to 5 bars, with a decreasing trend. Additionally, after 10 ns, the impact of IP on the simulated specimen's length change seems to diminish. When raising the IP from 1 bar to 5 bar, the atomic length falls from 11 to 10 Angstroms [34].

In other work, to improve the dispersion of graphene oxide and the compatibility of polypropylene with ethylene-propylene rubber (EPDM), polypropylene (PP) was strengthened by employing graphene oxide. Next, a PP-g-MA compatibilizer was utilized [35]. The aim of that research was to obtain a better blend composition. In this respect, utilizing TGA, the mechanical properties of the specimens were evaluated, and their theory was

studied. The findings show that raising the amount of graphene oxide with the addition of the PP-g-MA compatibilizer greatly improves the mechanical characteristics, such as tensile strength, elongation at break, and Young's modulus. Additionally, SEM images showed that the particle size of the EPDM dispersion phase decreases as the concentration of nanofillers in the mix matrix increases [35].

Recent works related to graphene-based polymer nanocomposites are summarized in Table 1.

**Table 1.** Summary of recent works related to graphene-based polymer nanocomposites.

| Sr. | Polymer | Percentage | Technique | Improvement | Reference |
|---|---|---|---|---|---|
| | | | **Graphene-Based Nanocomposites** | | |
| 1 | PMMA | 1 wt.% | Bulk polymerization | Thermal properties: 126% | [21] |
| 2 | PVA | 10 wt.% | Magnetic agitation | Thermal properties: 25% | [22] |
| 3 | MFC | 0.6 wt.% | Mixing | Thermal properties: 5% Mechanical properties: 23% | [23] |
| 4 | PF | 0.12 wt.% | Mechanical stirring | Thermal properties: 12% Mechanical properties: 150% | [24] |
| 5 | CMC | 2 wt.% | Ultrasonication | Mechanical properties: 132% | [25] |
| 6 | PVDF | 2 wt.% | Stirring | N/A | [26] |
| 7 | Epoxy | 4 wt.% | Sonication | Thermal properties: 43% | [27] |
| 8 | PET/PBT | 1 wt.% | Mixing | Thermal properties: 33% Mechanical properties: 17% | [28] |
| 9 | PU | 1 wt.% | Sonication | Thermal properties: 63% Mechanical properties: 58% | [29] |
| 10 | PU | 2 wt.% | Mechanical stirring | Thermal properties: 54.5% | [33] |
| 11 | DETDA and DGEBA | 1 to 5 wt.% | Mechanical stirring | Mechanical properties: 22% | [34] |

## 3. MXene-Based Nanocomposites

MXenes are newly identified 2D nanomaterials with exceptional mechanical, thermal, and tribological characteristics that are often used in a distinct range of crucial applications, including cancer therapy, energy, and environmental uses, owing to their unique characteristics, which include a mechanochemical origin with outstanding mechanical efficiency, heat resistance, and surface treatments. Thanks to strong interactions between macromolecules and the terminal units of 2D MXenes, they offer enormous promise in advanced polymeric composites [36].

MXene-reinforced polymer nanocomposites were fabricated to increase the mechanical and thermal properties of the MXene/polymer. For instance, to fabricate fiber-reinforced composites, thermosetting epoxy polymers are frequently employed as matrices because of their outstanding stiffness and strength [37]. Thermosetting epoxy has poor fracture toughness and is often brittle, which limits its uses. The use of micro- and nanoscale particles is one method for enhancing epoxy's mechanical characteristics. Due to their superior thermal and mechanical characteristics, MXenes, a broad family of 2D transition metals, can be employed to fabricate multifunctional nanocomposites. With MXene–epoxy composites, it was determined that the combining strength and microscopic mechanisms of breakage are under uni-axial stress. The MXene type ($Ti_2CTx$ or $Ti_3C_2Tx$) has little impact on the compelling powers of MXene–epoxy. Less hydrogen covering the $Ti_3C_2Tx$ surface results in greater binding between the MXene and epoxy because of its compatible nature. The stress transmission between the polymer and the particle causes the composites' 16% higher Young's modulus compared to pure epoxy; the modulus rises with the filler content up to 1 wt.% (Figure 3a). Because of filler aggregation, the rise in the modulus is smaller

with greater filler levels. Experimental studies of surface damage showed that spaces developed around the edges of the filler in MXene–epoxy during strain [37].

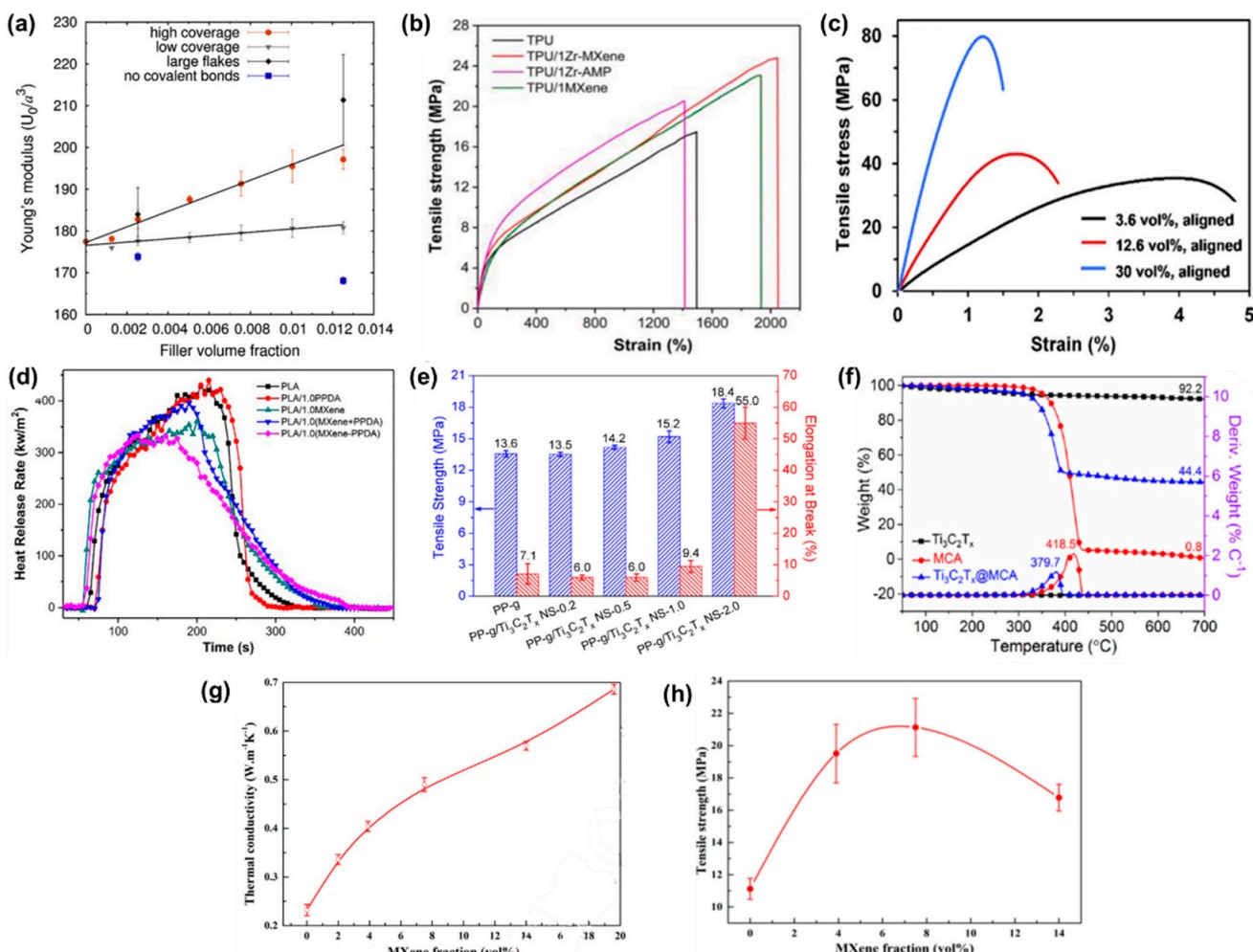

**Figure 3.** The mechanical properties of MXene–based nanocomposites: (**a**) Young's modulus of MXene–epoxy nanocomposite with different loadings of MXene [37], (**b**) tensile strength of MXene–TPU nanocomposites [38], (**c**) stress–strain curve of MXene–epoxy-based nanocomposite [39]. (**d**) Thermal stability of PlA/MXene nanocomposites [40]. (**e**) The tensile strength and elongation of pP/MXene nanocomposites [41]. Thermal properties of (**f**) TPU/$Ti_3C_2Tx$@MCA [42], (**g**) thermal conductivity with varying MXene contents [43]. (**h**) Tensile strength with different loadings of MXene [43].

A hybridized fire retardant (zr-MXene) was synthesized by in situ stacking zirconium amino-tris-(methylene phosphonate) (Zr-AMP) onto the surface of titanium carbide (MXene) [38]. In the second step, a nanocomposite of thermoplastic polyurethane (TPU)/zr-MXene was fabricated to observe the mechanical properties. According to the findings, adding 1 wt.% zr-MXene to TPU resulted in 2060% higher strain and a toughness increase of 316 MJ·m$^3$, the highest ever recorded, as well as 43.4% higher tensile strength (Figure 3b) and better fatigue compared to clean TPU. The combined physical and catalytic reactions of zr-MXene result in a TPU material that is much less flammable [38].

In another study, a nanocomposite of MXene/PVA/epoxy was fabricated to analyze the tensile strength and degradation performance [39]. Finite element modeling was used to calibrate all parameters, including the particle size, mechanical characteristics, and interface layer strength. The ultimate mechanical characteristics of representative volume element models and the MXene flake orientation were examined. According to the findings,

the Young's modulus and tensile strength of the nanocomposite are greatly improved by aligned and high-aspect-ratio nanoparticles. A model with 30 vol.% oriented MXene particles increases Young's modulus and tensile strength by 745% and 92%, respectively, relative to clean epoxy (Figure 3c) [39].

Moreover, phenyl phosphonic diaminohexane (PPDa)/MXene was fabricated by intercalating PPDA into the MXene interlayer. The MXene nanoparticle interlayer spacing was increased, and as-produced MXene-PPDA was evenly dispersed throughout polylactic acid (PLA) [40]. PLA reached a UL-94 V-0 rating via 1.0 wt.% MXene-PPDA, which resulted in a reduction of 23% in the peak heat release rate, showing increased flame resistance. In addition, 1 wt.% MXene-PPDA raises the PLA composite's primary degradation heat, which results in a 25-fold gain in yield compared to clean PLA. Furthermore, PLA's resilience is improved while maintaining its mechanical strength thanks to MXene-PPDA. The novel approach for the proposal of ingredients and the fabrication of high-tech polymer nanocomposites resulted in higher stability (Figure 3d), 190% higher mechanical strength, and low flammability [40].

Ultrathin 2D titanium carbide ($Ti_3C_2Tx$)/polypropylene (PP) drastically improved the tensile strength by 35.3% and increased deformability by 674.6%, as shown in Figure 3e [41]. Also, the initial decomposition temperature of the film was improved by 79.1 °C, and the modulus was enhanced by 102.2% in oxygen-free fast evaporation. A simple method for designing ductile and thermally stable composites was influenced by nano-reinforcement, as well as a model for expanding the use of 2D MXenes in matrices [41]. By modifying the surface of titanium carbide nanosheets ($Ti_3C_2Tx$, MXene) with melamine cyanurate (MCA) through physical bonding, the synthesis of a versatile nanohybrid, $Ti_3C_2Tx$@MCA, and the following TPU/$Ti_3C_2Tx$@MCA was studied. The resulting nanocomposite had a tensile strength of 62 MPa, toughness of 176 7.9 MJ·m$^3$, strain at break of 590%, and a 40% decrease in the temperature release rate. It also had a significant toughness of 3.0 wt.% $Ti_3C_2Tx$@MCA. Such exceptional mechanical and flame-resistant capabilities surpass those of its earlier competitors (Figure 3f) [42].

In another study, a nanocomposite of nitrile butadiene rubber (NBR)/$Ti_3C_2tx$ MXene was fabricated to analyze the thermal and mechanical properties of the prepared polymer nanocomposite [43]. Exfoliation in the elastomer was made possible by combining lithium ions and rubber into the MXene gap, which was discovered using XRD. The thermal conductivity of the nanocomposite was enhanced to 0.69 W·m$^{-1}$·K$^{-1}$, with a rise in Mxene contents, as shown in Figure 3g. A swelling ratio of 1.61 was reached at just 2.8 vol.% MXene, which is a 75% decrease from NBR incorporating either graphene or CNT particle content. Tensile properties and Young's modulus were improved as the MXene concentration was increased to the optimal level, showing 105% improvement (Figure 3h) [43].

An anti-freezing MXene nanocomposite organohydrogel (MNOH) was fabricated by immersing an MXene nanoscale hydrogel (MNH) in ethylene glycol (EG) [44]. The MXene nanosheet structures were incorporated into the hydrogel to produce the MNH. The MNOH is extraordinary in its anti-freezing capacity (−40 °C) and its high moisture-retaining stability (eight days), and it has high mechanical properties, such as 350% higher strain with the addition of a suitable amount of MXene [44].

The performance of MXenes makes them interesting prospects as reinforcements for polymer nanocomposites [45]. The existence of groups that were connected to the preparation used had a considerable impact on the mechanical properties. Compared to pure TPU, the tensile strength of the TPU/MXene nanocomposite was improved by 47% with the inclusion of 0.5 wt.% MXene. The interface between MXenes with various functionalities and TPU-based compounds was studied via density functional theory. It was noted that hydrogen bond formation and π-π stacking were the primary interaction processes that play an important role. Although fluorine and hydroxyl terminations promoted the interaction with TPU, oxygen-terminated MXene inhibited it. These results show the value of modifying the surface chemistry of MXenes to enhance MXene/polymer matrix connections in nanocomposites [45].

Similarly, natural rubber (NR) as the matrix and $Ti_3C_2tx$ MXene as the nano-reinforcement were used to fabricate nR/MXene nanocomposites for the study of mechanical properties [46]. At lower MXene concentrations, the electrostatic repulsive force is caused by the negative charges of MXene, and NR allows the MXene to disperse on the surface. These provide an interconnection for effective electron transport and load transfer. The strong 3D MXene network significantly reinforces the NR matrix, showing a 700% and 15,000% higher tensile strength and modulus, respectively, than those of pure NR [47]. In another study, a mechanically robust $Ti_3C_2Tx$ composite sheet obtained by mixing poly(3,4-ethylenedioxythiophene) with poly(styrene sulfonate) (PEDOT/PSS) was fabricated by removing the insulated PSS via concentrated $H_2SO_4$. The tensile strength was increased from that of the $Ti_3C_2Tx$ sheet to 38.5 2.9 MPa by adding 30 wt.% PEDOT–PSS and following up with an acidic process. The enhancement in tensile strength was up to 155% compared to that of $Ti_3C_2Tx$ films [46].

In other work, $Ti_3C_2tx$ MXene/silicone rubber (SR) nanocomposites were fabricated. The $Ti_3C_2tx$ MXene nanosheets' impacts on the mechanical and thermal properties of the nanocomposites were examined. The nanocomposites' thermal stability allowed them to withstand temperatures of 450 °C. In the nanocomposite containing 2 wt.% $Ti_3C_2tx$ MXene, a tensile strength of 430 kPa, an elongation at break of 341%, and a decreased elastic modulus of 402 kPa were also attained [48].

The characteristics of $Ti_3C_2Tx$/PVA nanocomposites are enhanced by the application of innovative intumescent flame retardants such as poly(vinylphosphonic acid) (PVPA) and polyethylenepolyamine (PEPA) [49]. The mechanical, thermal, and flame-retardant characteristics of MXene/PVA nanocomposites were tested. The findings demonstrate that MXene was uniformly distributed throughout the PVPA- and PEPA-containing PVA matrix. MXene/PVA nanocomposites' flame-retardant characteristics were significantly enhanced by PVPA and PEPA, but their thermal degradation was unaffected. Additionally, MXene increased the PVA matrix's thermal stability. Despite the presence of PVPA and PEPA in the PVA matrix, the elongation at break of MXene/PVA nanocomposites reached its maximum when the MXene mass fraction was 1.0 wt.%, while the tensile strength and Young's modulus were enhanced by 25.3% and 64.45%, respectively, with the rise in MXene content in the PVA matrix [49].

Recent works related to MXene-based polymer nanocomposites are summarized in Table 2.

**Table 2.** Summary of recent works related to MXene-based polymer nanocomposites.

| Sr. | Polymer | Percentage | Technique | Improvement | Reference |
|-----|---------|-----------|-----------|-------------|-----------|
| | | | **MXene-Based Nanocomposites** | | |
| 1 | Epoxy | 1 wt.% | Sonication | Mechanical properties: 16% | [37] |
| 2 | TPU | 1 wt.% | Stirring | Mechanical properties: 43.4% | [38] |
| 3 | PVA/epoxy | 10 wt.% | Probe Sonication | Mechanical properties: 92% | [39] |
| 4 | PLA | 1 wt.% | Mixing | Thermal properties: 23% Mechanical properties: 190% | [40] |
| 5 | PP | 2 wt.% | Mechanical stirring | Thermal properties: 79 °C Mechanical properties: 35.3% | [41] |
| 6 | TPU | 3 wt.% | Stirring | Thermal properties: 40% Mechanical properties: 62 MPa | [42] |
| 7 | NBR | 2.8 vol.% | Mechanical Stirring | Thermal properties: 180% Mechanical properties: 105% | [43] |
| 8 | TPU | 0.5 wt.% | Mixing | Mechanical properties: 47% | [45] |

**Table 2.** *Cont.*

| Sr. | Polymer | Percentage | Technique | Improvement | Reference |
|---|---|---|---|---|---|
| | | | **MXene-Based Nanocomposites** | | |
| 9 | NR | 6.71 vol.% | Mixing | Mechanical properties: 155% | [46] |
| 10 | SR | 2 wt.% | Mixing | N/A | [48] |
| 11 | PVA | 1 wt.% | Mixing | Mechanical properties: 23.5% | [49] |

## 4. Carbon-Nanotube-Based Nanocomposites

For the fabrication of nanocomposites, CNTs are suitable reinforcing materials, specifically for one-dimensional applications, such as fibers/wires. CNTs must be functionalized to improve their characteristics; nevertheless, due to their chemical inertness, complete dispersion is not possible. For instance, the elastic characteristics of SWCNTs and multi-walled carbon nanotubes were tested after functionalization with ethylene-diamine. Using CNTs with different densities of the linked ethylene-diamine molecules, molecular dynamics simulations were conducted. The impact of amine functionalization on the Young's modulus, shear, and tensile strength of various CNT structures was statistically investigated [50].

The influence of the interface on the mechanical properties of polymer-based composites was examined [51]. The mechanical characteristics of nanocomposites were analyzed using a closed micromechanical interphase model. This model incorporated the diameter of the CNTs, the thickness of the interphase, and the mechanical properties of the CNTs/polymer. The findings show that the interphase had a substantial impact on improving the elastic modulus of the CNTS/polymer, which increased by up to 108% with the optimal addition of CNTs [51].

Epoxy/CNT/carbon fiber nanocomposites were fabricated to analyze the stretch properties of composites [52]. CNT contents ranging from 0% as a reference to 0.5%, 1%, 2%, and 4% were utilized. There were several mechanical tests, including an open-hole tension test, a shear beam experiment, and flatwise tension studies. In the developed composites, the start of deterioration and the spread of fractures were managed. According to the findings, adding 2 wt.% CNTs increased the composite's mechanical efficiency by up to 70%, as illustrated in Figure 4a [52].

There is a suitable enhancement of the mechanical properties when using CNT-reinforced polymer nanocomposites [53]. The distribution, aggregation, and net characteristics of the polymer, as well as the structural defects and covalent bonding formed during the functionalization process, all affect the mechanical performance of the CNT/polymer nanocomposite (CNTPN). A multiscale approach was developed whereby the distribution and aggregation of CNTs were explored on a microscale, while their structure and interfacial phase were investigated in a nanoscale simulation. Based on the parameters of the pristine polymer, the dispersion and clustering of CNTs can result in an improvement in the strength of the CNTPN [53]. Additionally, the ultimate strength of the CNTPN is raised by 34% during the functionalization process of CNTs, as revealed in Figure 4b. However, Young's modulus of the nanocomposite decreases when structural defects in CNTs that were produced during the functionalization process are present. Raising the curvature of CNTs also greatly reduces the functionalization process's beneficial effects, and the final strength of the functionalized CNT polymer nanocomposite (FCNTPN) is comparable to that of the intact CNT polymer nanocomposite (ICNTPN). The aggregation of CNTs alters the fracture process and has a negative impact on Young's modulus [53].

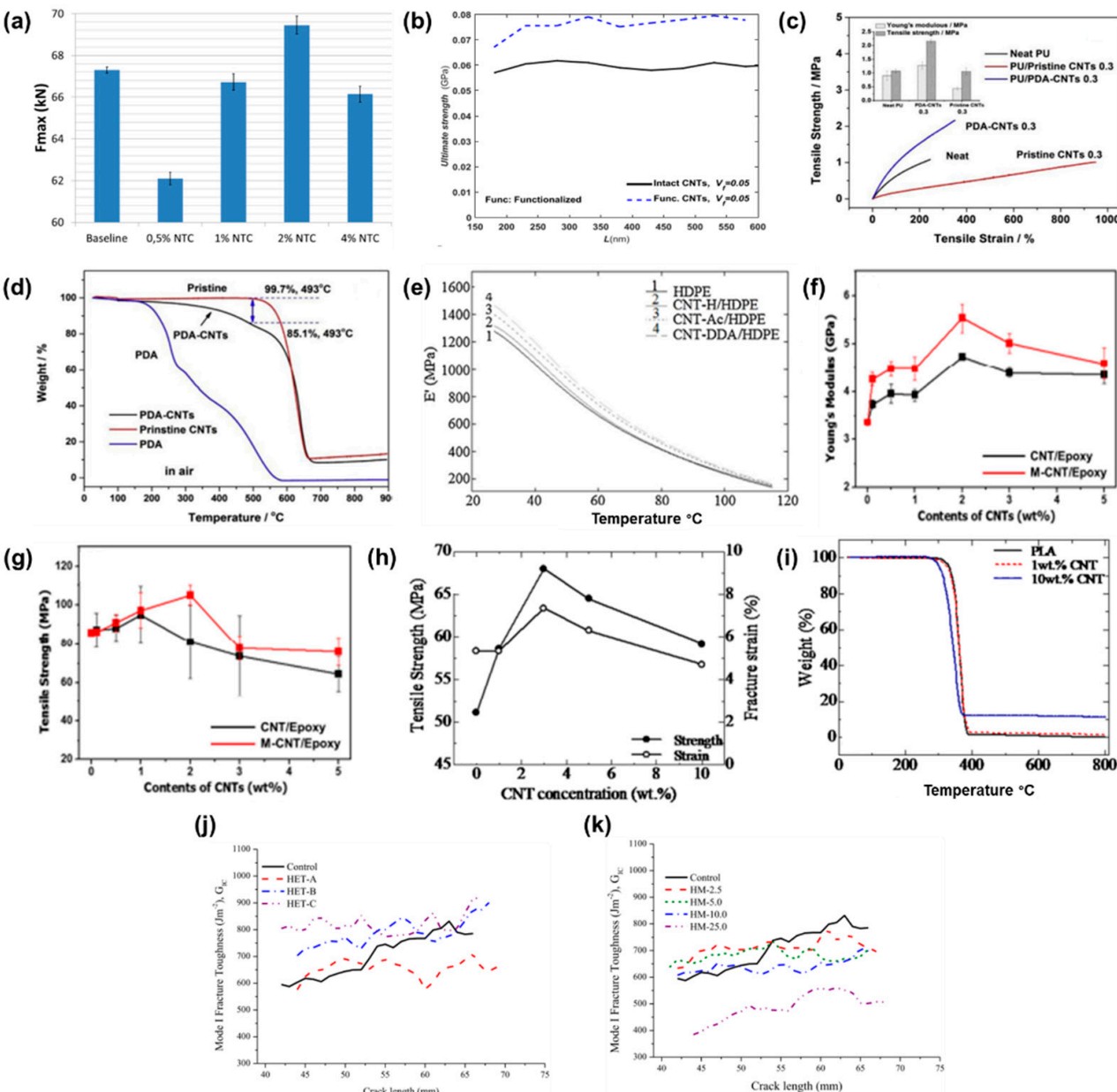

**Figure 4.** Mechanical properties: (**a**) strength of CNT-epoxy-based CFRPC at various loadings of CNTs [52], (**b**) ultimate strength of functionalized CNT-based polymer composites [53], (**c**) stress–strain curves of pure PU, PU/CNTs, and PU/PDA-CNTs [54]. (**d**) Thermal properties of pure PU/CNTs and PU/PDA-CNTs [54]. (**e**) Storage modulus of CNT-based polymer composite [55]. Mechanical properties of CNTs/epoxy: (**f**) Young's modulus and (**g**) tensile strength [56]. (**h**) The tensile strength and fracture strain and (**i**) TGA of pure PLA and PLA/CNTs [57]. Mode I fracture toughness of CNT-based polymer composites (**j**) for homogeneous dispersion and (**k**) for heterogeneous dispersion [58].

Using the solution-casting approach and in situ polymerization, a nanocomposite containing modified polydopamine-CNTs (PDA-CNTs) was dispersed in a PU matrix to fabricate PU/PDA-CNTs [54]. The tensile strength and modulus of the nanocomposite were increased by 40% and 100%, respectively, with a loading of 0.3 wt.% PDA-CNTs (Figure 4c). The incredibly improved interfacial interactions between the matrix and the reinforcement were studied. Adding PDA-CNTs significantly changed the phase separation in PU, forming a special network-like structure that produced the unanticipated reinforcing

effect. The thermal properties of the PU coating were enhanced by 15% with the addition of the optimal content of PDA-CNTs, as shown in Figure 4d [54].

The impact of functionalizing CNTs with dodecylamine (DDA) on the thermal and mechanical properties of composites based on HDPE was studied [55]. Therefore, the CNTs' dispersion and interfacial adhesion with HDPE were both enhanced by surface modification with DDA via non-covalent bonding. The thermal and mechanical properties were increased by 15% (Figure 4e), which was related to the enhanced CNT dispersion and interface in the HDPE substrate as a factor of its hydrophilicity [55].

The mechanical characteristics of epoxy/CNT nanocomposites were improved with the inclusion of CNTs, which serve as nanofillers [56]. The CNTs were functionalized with melamine to enhance the interfacial strength between the CNTs and epoxy. Using CNT/epoxy and melamine-CNT (M-CNT)/epoxy nanocomposites, tensile testing and single-edge notch bending were conducted at different loadings of CNTs. The M-CNT/epoxy showed improvements in Young's modulus of 64% and ultimate tensile strength of 22% with the inclusion of 2 wt.% M-CNTs (Figure 4f,g). Additionally, it was shown that at 2 wt.% M-CNTs, the nanocomposite significantly increased the fracture toughness by 95%. The improvements in the modulus and strength were examined and linked to the uniformity of CNTs in epoxy. Investigations on the phenomenon of crack propagation were made in relation to increasing the fracture toughness [56].

A CNT/PLA nanocomposite was fabricated using a twin-screw extrusion machine with different CNT contents to analyze the performance of the nanocomposite [57]. The influence of CNT contents on the mechanical and thermal properties of the CNTs/PLA was observed, as shown in Figure 4h,i. These findings revealed that the crystallinity increased as CNT contents increased, indicating that CNTs served as a nucleating agent in PLA. Furthermore, at 3 wt.% CNT inclusion and effective CNT distribution and dispersion, the tensile strength increased by 32.12%, and thermal characteristics were enhanced by 2.8-fold at 10 wt.% CNTs [57].

A wet powder impregnation approach was employed to prepare fiber-reinforced hierarchical composites that were subsequently embedded with 25 wt.% CNTs [58]. To create CNT-rich zones with spatial separation, microstructural heterogeneity in the matrix of these laminates was generated by wet powder impregnation. In comparison to the baseline carbon-fiber-reinforced polymer nanocomposites (CFPNCs) and hierarchical composites with homogeneously dispersed CNTs all around the matrix with identical CNT loading, the Mode I fracture toughness of these heterogeneous hierarchical composites was enhanced by 41% and 26%, respectively. This enhancement in fracture toughness was due to a tortuous crack path, as shown in Figure 4j,k. The matrix microstructural heterogeneity had little effect on the interlaminar shear strength [58].

High-density polyethylene (HDPE) composites reinforced with CNTs were developed by melting the HDPE [59]. The CNT content and sonication temperature during specimen preparation were important considerations in the primary composite design analysis of the mechanical characteristics. Using 0.8 wt.% CNT content and a sonication temperature of 55 °C, the nanocomposite exhibited 43% higher hardness when compared to the pure polymer. Improvements in the nanocomposite's thermophysical and viscoelastic characteristics are responsible for the enhancement in mechanical characteristics [59].

In another study, a polymer nanocomposite (PNC) of PP/CNTs was fabricated to analyze the effect of CNTs on the thermal performance of PNCs [60]. The thermal performance of PNCs incorporating CNTs was originally improved by microwave radiation. Low CNT loading rates in PP nanocomposites led to high thermal resistance. The CNTs were heated by converting electromagnetic energy into mechanical vibrations while being protected by an inert atmosphere. The heat produced by the CNTs easily melted the surface of PP, and after a sufficient microwave treatment duration and subsequent hot pressing, a uniform CNT network was created. The crystallization temperature of both neat PP and PP/CNT PNCs (12% higher) was affected by the optimal loading of CNTs and processing temperature [60].

A study of the tensile strength of CNTPN was performed using the Ouali technique, considering network development beyond the percolation threshold [61]. The proposed approach also considers the connecting and reinforcing functions of the interphase since interphase zones frequently arise in nanocomposites. To observe the mechanical properties of the polymer matrix, nanoparticles, and interphase, several equations were formed. The developed model exhibits a suitable level of consistency with the nanocomposite's findings. The proposed model also relates various variables, such as the radius and vol.% of CNTs and strength, as well as the volume fraction, percolation threshold, and strength of the interphase. These findings support the established model's capacity to analyze the tensile strength of CNTPN [61].

To assess its impact on the thermal and mechanical characteristics of 3D-printed PLA/CNT nanocomposites, CNT functionalization was carried out using a $HNO_3$ solution to introduce defects and oxygen functional groups onto the CNT surface [62]. The findings showed that, in contrast to the usage of conventional CNT (c-CNT), functionalized CNTs (f-CNT) demonstrated superior dispersion in the matrix and functioned as more effective nucleating agents for PLA crystallization. The mechanical strength of the 3D-printed parts was significantly increased from 29.4 0.7 MPa for PLA/c-CNT to 41.6 1.4 MPa for PLA/f-CNT with the addition of just 0.5 wt.% f-CNT, and the interfacial adhesion between the 3D-printed layers was improved, maintaining the thermal stability of the nanocomposites. When f-CNT was utilized as reinforcement, dynamic mechanical investigations showed a considerable increase of 43% in the storage modulus at 37 °C. As a result, the addition of f-CNT considerably enhanced the thermal and mechanical characteristics of 3D-printed PLA/CNT nanocomposites [62].

In contrast to SWCNTs, a nanocomposite of MWCNTs/polymer matrix was fabricated to analyze the thermal and mechanical properties. For instance, an epoxy/MWCNT nanocomposite was fabricated to study the possible mechanical properties of epoxy resin [63]. It was observed that increasing the content of MWCNTs enhanced the flow stress and fracture strain. The stress flow and fracture strain were increased by 37% and 50%, respectively, with the inclusion of 0.5 wt.% MWCNTs. More crystalline regions were observed in the MWCNT composite, resulting in a stronger composite [63].

To increase the interfacial adhesion between polymer and nanotubes, the surface of MWCNTs was chemically functionalized [64]. The ability to improve the various physical and chemical characteristics of MWCNTs through functionalization was effective in improving the dispersion and interfacial strength. The influence of the functionalized filler loading in the MWCNT-embedded PMMA nanocomposite on a variety of mechanical properties was investigated. In Figure 5a, it is shown that the tensile properties of the PMMA/MWCNT nanocomposite were enhanced by 16% with the incorporation of 0.5 wt.% MWCNTs [64].

As a low-temperature synthesis method, graphitic structure design (GSD) was used to fabricate CFRPs by growing MWCNTs on carbon fiber fabrics [65]. There are two distinct morphologies of MWCNT forests: uniform and patterned. The fabrication of CFRPs depends on hybrid reinforcement. A double cantilever beam test confirmed that surface-grown nanoparticles affected the Mode I interlaminar fracture toughness (GIc) of CFRPs. As a result of uniform and checkerboard-patterned growth morphologies, the surface-grown MWCNTs increased the GIc of the CFRPs by 22 and 32%, respectively (Figure 5b) [65].

The effects of MWCNTs on the flexural and high-velocity performance of fiber metal laminates (FMLs) composed of basalt fibers, epoxy, and aluminum 2024-T3 were studied [66]. The addition of the MWCNTs had a substantial impact on the adhesion of composite piles and the interfaces between aluminum and basalt fibers/epoxy. The strong adherence of MWCNTs to the FMLs significantly enhanced the flexural characteristics. As shown in Figure 5c,d, the flexural strength and flexural modulus were increased by 36.62% and 60.16%, respectively, with the addition of 0.5 wt.% MWCNTs [66].

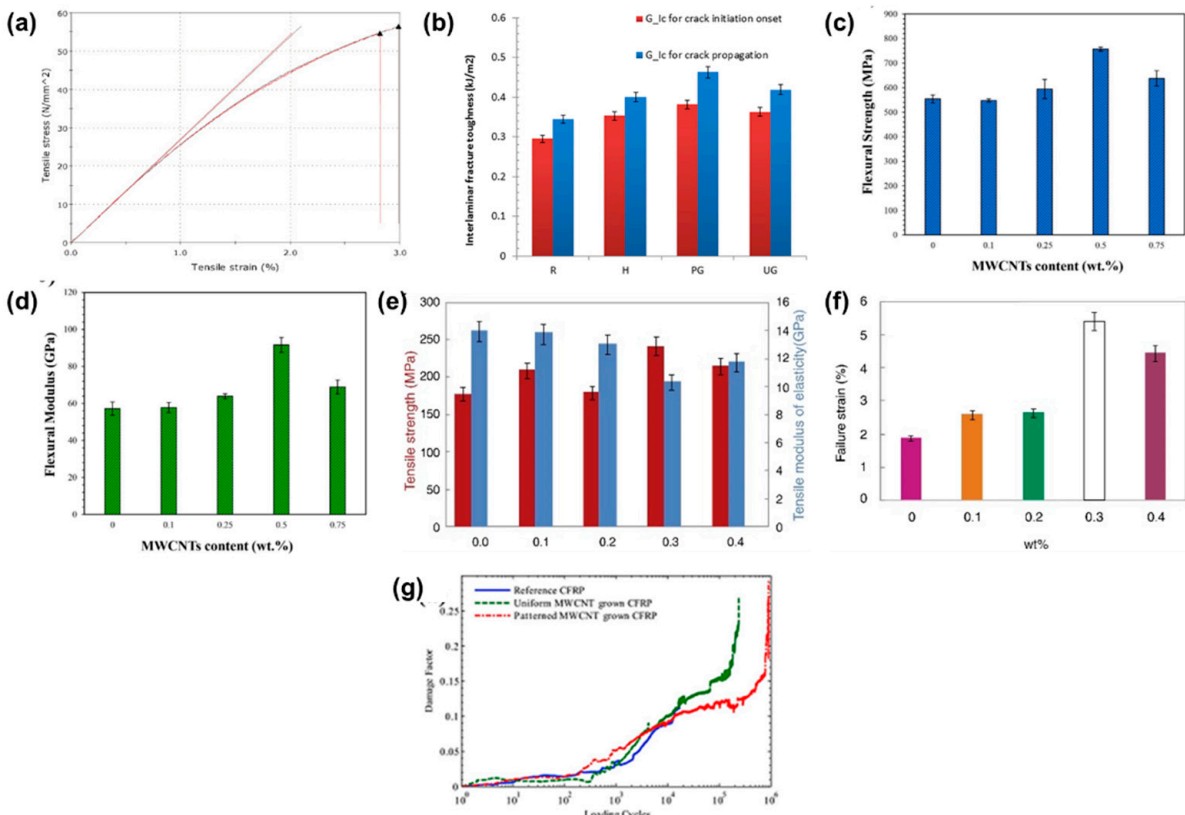

**Figure 5.** The mechanical properties of MWCNT-reinforced polymer composites: (**a**) stress–strain curves of MWCNTs/PMMA [64], (**b**) interlaminar fracture toughness of MWCNT-reinforced CFRPs [65], (**c**) flexural strength of MWCNTs/epoxy [66], (**d**) flexural modulus of MWCNTs/epoxy [66], (**e**) tensile properties of MWCNT-reinforced GFRPs [67], (**f**) failure strain of MWCNT-reinforced GFRPs at different concentrations of MWCNTs as represented by different colors [67], and (**g**) tension–tension fatigue loading of MWCNT-reinforced CFRP [68].

Glass-fiber-reinforced polymer composites (GFRPs) with low MWCNT concentrations were fabricated and mechanically characterized [67]. The influence of different concentrations of MWCNTs on the mechanical characteristics of GFRPs was explored. In comparison to pure GFRPs, the composite of GFRPs embedded with 0.3% MWCNTs exhibited 243 MPa greater tensile strength and 332.53 MPa flexural strength, as shown in Figure 5e,f. MWCNTs are crucial for improving mechanical properties, including tensile strength, failure strain, and hardness [67].

Homogeneous and heterogeneous MWCNTs were synthesized over carbon fibers using a comparatively non-destructive manufacturing process [68]. CFRPs based on surface-synthesized MWCNTs were manufactured and tested under tensile testing. The composite was also determined under tension–tension fatigue loading at various stress levels of 85, 90, and 95% of the strength (Figure 5g) under a cyclic stress ratio of 0.10 force Newton. The impact of different topologies of the surface-grown MWCNTs on the fatigue damage formation and life of the CFRPs was analyzed. The outcome shows that the fatigue life of CFRPs was enhanced by 150% with the patterned development of MWCNTs [68].

In order to increase the interlaminar shear and flexural strength of the laminates, a novel composite was fabricated with the inclusion of MWCNTs in the polymer [69]. The flexural and interlaminar shear strength of CFRPs were enhanced by 24% and 28% in comparison to the pristine epoxy, respectively. According to the findings, the delamination factor decreased by 21% and 28.60%, respectively, on the start and exit sides [69].

Similarly, a TPU/MWCNT nanocomposite was fabricated using the fused deposition method to analyze the mechanical properties of sensors under cyclic loadings [70]. The elastic modulus of TPU/MWCNTs showed a negligible reduction, i.e., 14%, compared to that of

bulk equivalents, suggesting good interlayer adhesion and enhanced properties. As a result, through-layer and cross-layer conductivities were essentially maintained upon printing. At applied stresses of up to 100%, piezoresistivity gauge values as high as 176% were reached. Against cyclic loadings, a very reproducible resistance–strain behavior was also observed. The findings show that TPU/MWCNT is a suitable piezoresistive substrate for 3D printing, with possible advantages in prosthetics, soft robotics, and wearable electronics, where complicated design, multi-directionality, and customizability are required [70].

Epoxy resin was used to create hybrid polymer nanocomposites that were reinforced with MWCNTs and zirconium dioxide ($ZrO_2$) and yttrium oxide ($Y_2O_3$) nanoparticles (NPs). The hybrid nanocomposites' mechanical and thermal stabilities were thoroughly examined [71]. It was demonstrated that the mechanical and thermal characteristics of hybrid nanocomposites may be enhanced by the inclusion of a small amount of MWCNTs. For example, the hybrid nanocomposite had a maximum tensile strength of 48.92 MPs and Young's modulus of 2492.06 MPa, respectively, corresponding to increases of 23% and 37% in comparison to plain epoxy when it contained 0.1 wt.% of MWCNT [71].

Recent works related to carbon-nanotube-based polymer nanocomposites are summarized in Table 3.

**Table 3.** Summary of recent works related to carbon-nanotube-based polymer nanocomposites.

| Sr. | Polymer | Percentage | Technique | Improvement | Reference |
|-----|---------|-----------|-----------|-------------|-----------|
| **Carbon-Nanotube-Based Nanocomposite** | | | | | |
| 1 | Epoxy | 2 wt.% | Mixing | Mechanical properties: 108% | [52] |
| 2 | PU | 0.3 wt.% | Mixing | Thermal properties: 15% Mechanical properties: 40% | [54] |
| 3 | HDPE | 0.8 wt.% | Stirring | Thermal properties: 15% Mechanical properties: 15% | [55] |
| 4 | Epoxy | 2 wt.% | Centrifugal mixing | Mechanical properties: 22% | [56] |
| 5 | PLA | 3 wt.% | Mixing | Thermal properties: 2.8 folds Mechanical properties: 32.12% | [57] |
| 6 | HDPE | 0.8 wt.% | Magnetic Stirring | Thermal properties: 55 °C Mechanical properties: 43% | [59] |
| 7 | PP | 1.3 wt.% | Microwave irradiation | Thermal properties: 12% | [60] |
| 8 | PLA | 0.5 wt.% | Mixing | Mechanical properties: 41.4% | [62] |
| 9 | Epoxy | 0.5 wt.% | Mixing | Mechanical properties: 50% | [63] |
| 10 | PMMA | 0.5 wt.% | Stirring | Mechanical properties: 16% | [64] |
| 11 | Epoxy | N/A | Mixing | Mechanical properties: 32% | [65] |
| 12 | Epoxy | 0.5 wt.% | Ultrasonication | Mechanical properties: 60.16% | [66] |
| 13 | Epoxy | 0.3 wt.% | Ultrasonication | Mechanical properties: 41.2% | [67] |
| 14 | Epoxy | N/A | Mixing | Mechanical properties: 150% | [68] |
| 15 | Epoxy | 1.5 wt.% | Sonication | Mechanical properties: 28% | [69] |
| 16 | TPU | 5 wt.% | Mixing | N/A | [70] |
| 17 | Epoxy | 0.1 wt.% | Mixing | Mechanical properties: 37% | [71] |

## 5. Carbon-Black-Based Nanocomposites

2CB is a carbon nanoparticle derived from the partial thermal decomposition of carbon-containing petroleum compounds. Due to its increased electrical conductivity, high surface area, and stability, CB is typically utilized as a nano-reinforcing material [72]. For instance, CB-reinforced polymer nanocomposites were studied to enhance thermal and mechanical properties. Great potential for increasing the fracture and post-cracking toughness of

CFRPs based on thermosetting epoxy was observed [73]. The hierarchy method served as inspiration for the synergistic incorporation of multiscale composites that included macro-fiber and nanoscale reinforcements. The CB surface was effectively changed for the nanoscale filler via ozone treatment, leading to extremely effective interfacial and dispersion characteristics. The enhancements in dispersion and interfacial characteristics should mostly be due to mechanical interlocking (Figure 6a). The optimal amount of ozone-functionalized CB in all the multiscale composites was around 5 wt.%, which increased the fracture and post-cracking toughness by 13% and 62%, respectively, compared to pure epoxy [73].

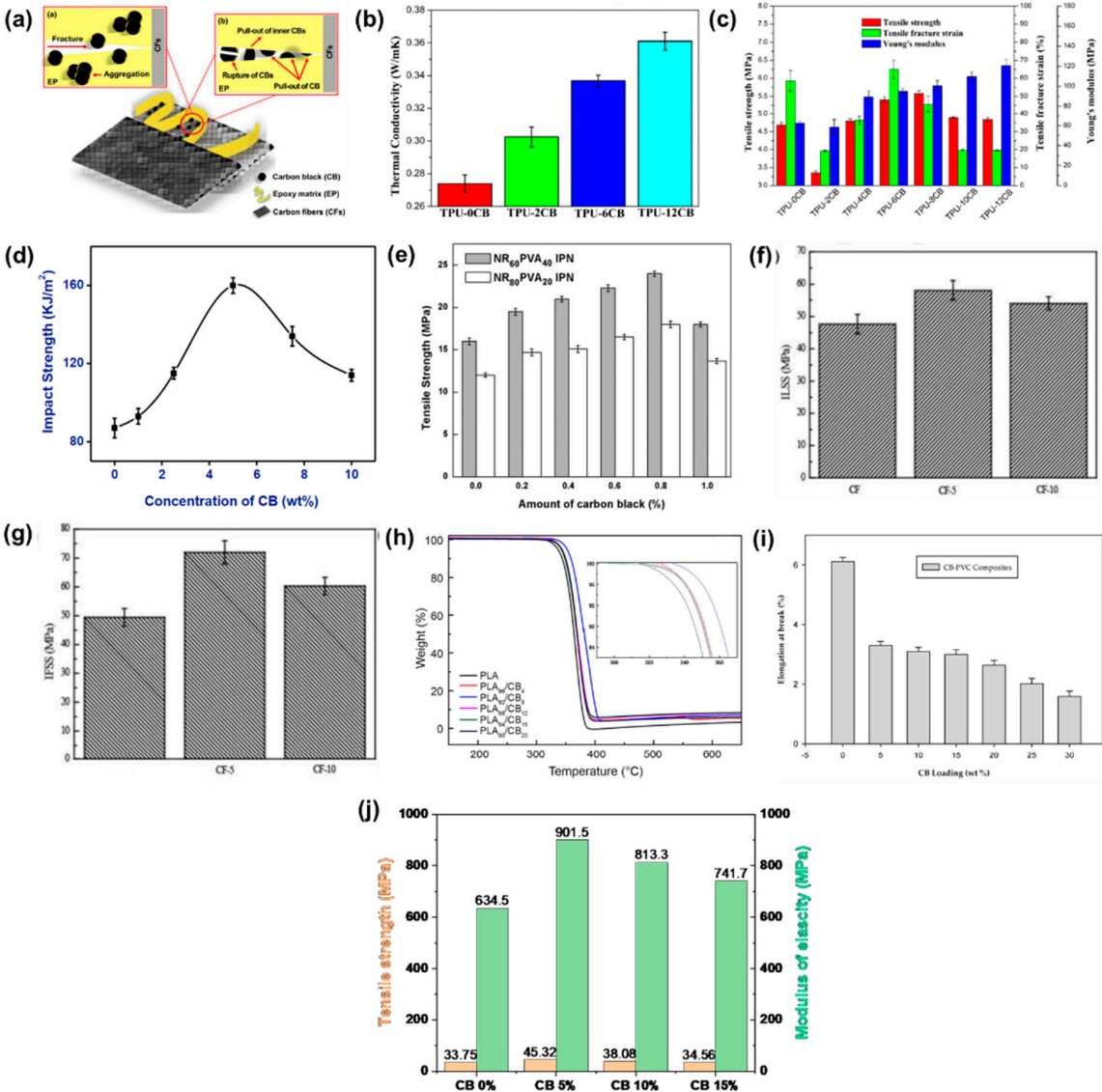

**Figure 6.** (**a**) Schematic representation of the mechanical interlocking mechanism of CB/epoxy [73], (**b**) thermal conductivity of TPU/CB composites [74], (**c**) tensile properties of TPU/CB at various loadings of CB [74], (**d**) impact strength of PP/CB [75], (**e**) tensile strength of PVA/NR/CB with various loadings CB [76]. Mechanical properties of CB-reinforced CFRPC: (**f**) ILSS and (**g**) IFSS [77]. (**h**) Thermal properties of PLA/CB with different contents of CB [78], (**i**) elongation of CB/PVC at various loadings of CB [79], and (**j**) illustrations of the tensile strength and modulus of elasticity of CB-reinforced epoxy composites [80].

By employing a combined co-coagulation and hot-pressing method, CB/TPU nanocomposites with different contents of nanoparticles were fabricated [74]. Hydrogen bonding exists between the CB and TPU polymers. Compared to neat TPU, the mechanical properties of TPU/CB nanocomposites were enhanced because of the uniform distribution of CB particles throughout the TPU matrix. As shown in Figure 6b,c, CB/TPU enhanced the thermal conductivity by 34% with the inclusion of 12% CB, and tensile strength was enhanced by 111% with the inclusion of 6% CB [74].

The mechanical properties of PP as a polymer, acrylonitrile–butadiene–styrene (ABS) as a rubber, and CB as a nanomaterial were studied [75]. The nanocomposite was fabricated utilizing an injection molding process and a dual-screw extruder. Due to its unique structure and qualities, CB has become a significant class of nanofillers with improved mechanical properties. The present polymer matrix is changed by adding nanofillers, which have more advantageous morphological and mechanical characteristics and may be used economically. In Figure 6d, it is shown that with CB at 5 wt.%, the impact strength of the nanocomposite was improved by 88% [75].

CB was added as reinforcement to the matrix of a completely interpenetrating polymer network (IPN) made up of NR and PVA to fabricate a nanocomposite [76]. Glutaraldehyde served as the common crosslinking agent for the components during the preparation of the IPN. Investigations were performed to observe the wt.% of CB that affected the composites' physical characteristics. By adding CB, a notable increase in the blend's mechanical strength and thermal stability was observed. The nanocomposite containing 0.8% CB increased the 50% tensile strength, as shown in Figure 6e. SEM images show that a higher concentration of CB led to the formation of agglomerates, which decreased the tensile characteristics [76].

The mechanical properties of CFRPs are significantly affected by the interfacial qualities between the CF and the matrix [77]. CB was added to the surface of CFs by chemical vapor deposition (CVD) to enhance the mechanical characteristics of fibers/epoxy without reducing the tensile strength of the basic fibers. The change in surface roughness and the distribution of CBs on the fiber surface were explored. The 5 min modified CFs showed considerable improvements in interlaminar shear strength (ILSS) and interface shear strength (IFSS), with values of 22% and 44%, as shown in Figure 6f,g, respectively. Additionally, in comparison to unmodified CFs, modified CFs demonstrated a slight increase in tensile strength [77].

The CB concentration affects the thermodynamic and mechanical characteristics of composites made of PLA and CB [78]. The findings demonstrated that incorporating CB at a suitable content enhanced the mechanical characteristics of PLA by 45%; however, introducing further CB content decreased the mechanical properties of PLA/CB. The cross-sections of PLA/CB became more wavy and coarser as the amount of filler increased from 4 to 12 wt.%. It suggested that the morphology of the composites was steadily altered and became more brittle. With CB at 8 wt.%, according to TGA, PLA's thermal stability increased by 15 °C, as shown in Figure 6h. The $T_g$ of PLA/CB was higher compared to pristine PLA, although the Tm was a little lower than that of pristine PLA, according to a DSC investigation. The developed composite material demonstrates exceptional characteristics because of its optimized CB concentration [78].

The nanofiller CB was embedded with thermoplastic polyvinyl chloride (PVC) by a compression molding process to create CB-reinforced PVC composites [79]. With CB loadings ranging from 5 to 30 wt.%, several kinds of CB-PVC compression-molded composites were fabricated. These composites' tensile characteristics were explored. As shown in Figure 6i, the elongation at break was reduced by 60% by the inclusion of 5 wt.% CB, while the tensile strength of the CB-PVC composites was increased to 33% by the incorporation of 15 wt.% CB [79].

The mechanical and thermal properties of CB-embedded epoxy resin nanocomposites were analyzed [80]. For this purpose, CB/epoxy resin nanocomposites were fabricated using hand layup and compression molding methods. The synthesized CB was nanoscale and evenly dispersed in the epoxy matrix, according to the morphological study. The

nanocomposite was thoroughly examined using mechanical and thermal testing. Epoxy with a CB content of 5 wt.% improved the tensile strength by 32% and the modulus of elasticity by 42%, as shown in Figure 6j. This composite is a possible option for polymer coatings in the automotive industry thanks to its higher tensile strength and hardness compared to pure epoxy [80].

The thermal and mechanical characteristics of vinyl ester/glass-reinforced composites with various concentrations of CB were tested [81]. The thermal resistance and surface fracture of the glass composite was tested using TGA and SEM, respectively. With increasing CB content, the degradation temperature gradually decreased. CB raised the glass transition temperature; however, over 500 °C, the weight loss of all specimens was similar. Compared to the other amount of CB, 4% CB with vinyl ester increased the tensile strength by 30%, hardness by 35%, flexural strength by 45%, flexural modulus by 66%, and interlaminate shear strength by 44% [81].

The impact of CB's content on the mechanical characteristics of ternary rubber nanocomposites made of NR, styrene-butadiene rubber (SBR), and NBR was explored [82]. To create the nanocomposites, a melt-mixing technique was used. The optimized specimen had greater tensile strength with a loading percent of 45 phr (parts per hundred of the rubber), offering better rigidity. Furthermore, the improved specimen's tensile strength increased by 233%. These findings demonstrate that the compatibility between CB and NR/SBR/NBR may be increased by using the optimal nanofiller loading [82].

The influence of the temperature and strain rate on the mechanical properties of various blend ratios of carbon black–butyl rubber (IIR)/high-molecular-weight polyethylene (PE) were investigated [83]. The motivation was to explain the brittleness phenomenon, which is strongly connected to all tensile mechanical parameters, including the blend strength and elasticity modulus. In particular, comparable dependence on the strain rate and temperature was shown for enhancing the starting Young's modulus by 447%, reducing the tensile strength by 60%, and decreasing the strain at failure by 61%. With a temperature rise, these characteristics decreased, and with a rise in the strain rate, these properties improved. In addition, except for Young's modulus in reverse, the tensile strength and strain at failure dropped for all temperature ranges when the PE percentage increased in the mixture [83].

Because of their superior thermal and mechanical properties, ethylene propylene diene monomer (EPDM)-based thermal insulators are an effective category of high-temperature thermal insulators [84]. To improve the elastomer's characteristics, CB was introduced, and the effect of various CB loadings was examined. The elongation, hardness, and tensile strength were all increased. A small improvement had also been made in the thermal stability, as measured by TGA. Compared to the pure specimen, the weight loss from the composites containing 10 phr was 28% lower [84].

A nanocomposite of CB/NR was fabricated using various types of CB-loaded nanocomposites to analyze the thermal performance [85]. It demonstrated that the primary cause of the increased heat conductivity of rubber composites with CB filler is the connective structure of CB. Because of the higher degree of bonding of CB in rubber, the improvement becomes more noticeable as the filler content increases. Additionally, CB aggregates with larger branch chains have better thermal conductivity and connections. To study the thermal conductivity of the composites, the Agari model was used. The Agari model predicts thermal conductivities with an average variation from the data of roughly 4%, which is lower than that of other traditional models [85].

Recent works related to carbon-black-based polymer nanocomposites are summarized in Table 4.

**Table 4.** Summary of recent works related to carbon-black-based polymer nanocomposites.

| Sr. | Polymer | Percentage | Technique | Improvement | Reference |
|---|---|---|---|---|---|
| | | | **Carbon-Black-Based Nanocomposites** | | |
| 1 | Epoxy | 5 wt.% | Stirring | Mechanical properties: 62% | [73] |
| 2 | TPU | 12 wt.% | Co-coagulation | Thermal properties: 34% Mechanical properties: 111% | [74] |
| 3 | PP/ABS | 5 wt.% | Mixing | Mechanical properties: 88% | [75] |
| 4 | PVA | 0.8 wt.% | Stirring | Mechanical properties: 50% | [76] |
| 5 | Epoxy | N/A | Mixing | Mechanical properties: 44% | [77] |
| 6 | PLA | 8 wt.% | Melt compounding | Thermal properties: 15 °C Mechanical properties: 45% | [78] |
| 7 | PVC | 15 wt.% | Mixing | Mechanical properties: 60% | [79] |
| 8 | Epoxy | 5 wt.% | Magnetic Stirring | Mechanical properties: 32% | [80] |
| 9 | Vinyl ester | 4 wt.% | Ultrasonication | Mechanical properties: 33% | [81] |
| 10 | NR/SBR/NBR | 45 parts per hundred of the rubber (phr) | Mixing | Mechanical properties: 233% | [82] |
| 11 | EPDM | 0.5 wt.% | Vulcanization | Thermal properties: 28% | [84] |
| 12 | NR | N/A | Vulcanization | Thermal properties: 4% | [85] |

## 6. Carbon-Quantum-Dot-Based Nanocomposites

CQDs are an emerging class of carbon nanomaterials that have gained considerable importance because of their distinctive characteristics. The water solubility, adaptability, photochemical properties, ease of surface modification, antibacterial activity, and superior biocompatibility of CQDs are evident benefits [86]. CQD/polymer nanocomposites have been fabricated via different methods to study their mechanical and thermal properties. For instance, the surface conductivity of a polyester composite reinforced with flax fiber and CQDs was examined through its mechanical and thermal properties [87]. Adding CQDs improved the polyester composite's properties and led to the fabrication of a sustainable material for cutting-edge applications. CQDs were synthesized by pyrolyzing sugarcane bagasse, and the composites were subsequently made by hand layup. The tensile, flexural, and impact strength of a polyester composite increased by 62%, 54%, and 938%, respectively, as the content of CQDs was raised by up to 5 vol.%. When the CQD concentration was increased to 7 vol.%, these characteristics of the nanocomposites were reduced. The polyester composite also demonstrated good heat transfer bridge formation with outstanding thermal conductivity (0.394 W/mK) [87].

The characteristics of smart composite coatings constructed from epoxy/PU reinforced with CQDs with multiple uses were synthesized [88]. This was accomplished by combining epoxy resin with a PU pre-polymer to create flexible and transparent polymer coatings. By adding 1–3 wt.% CQDs, luminous coatings were fabricated with increases in UV protection of 85–98%, increases in tensile strength of 10–35%, and improvements in viscoelastic characteristics, including storage modulus and $T_g$. According to the results of a salt spray test, coatings with higher CQD concentrations exhibit better anti-corrosive properties [88].

A unique electrospinning process was used to fabricate nanocomposite fibers made of CQDs coated with folic acid in a pectin/polyethylene oxide (PEO) matrix [89]. A one-step hydrothermal approach was used to synthesize CQDs that were nitrogen-doped and nitrogen/sulfur-codoped, with average diameters of 2.74 nm and 2.17 nm, respectively. The shape, physicochemical characteristics, and drug-release behavior of nanocomposite fibers containing CQDs were studied. According to the findings, CQDs increased the tensile strength of the fibrous scaffolds from 13.74 to 35.22 MPa while preserving similar flexibility.

The regulated drug release over 212 h and the biocompatibility of the nanocomposite fibrous scaffold make tissue regeneration highly feasible [89].

CQDs hydrothermally synthesized from paddy straw were used as nano-reinforcements in the fabrication of nanocomposites [90]. Stone waste particle nanocomposites reinforced with CQDs were fabricated and showed high strength, with an increase of up to 58% as compared to those without CQDs (Figure 7a). The CQD reinforcement allowed stone nanocomposites to attain high flexural strength of up to 60 MPa. The strong mechanical strength and sp$^2$-hybridized covalent links of CQDs and their high conductivity were ascribed to the high flexural strength and low dielectric constant of the CQD–stone waste polymer composite sheet [90].

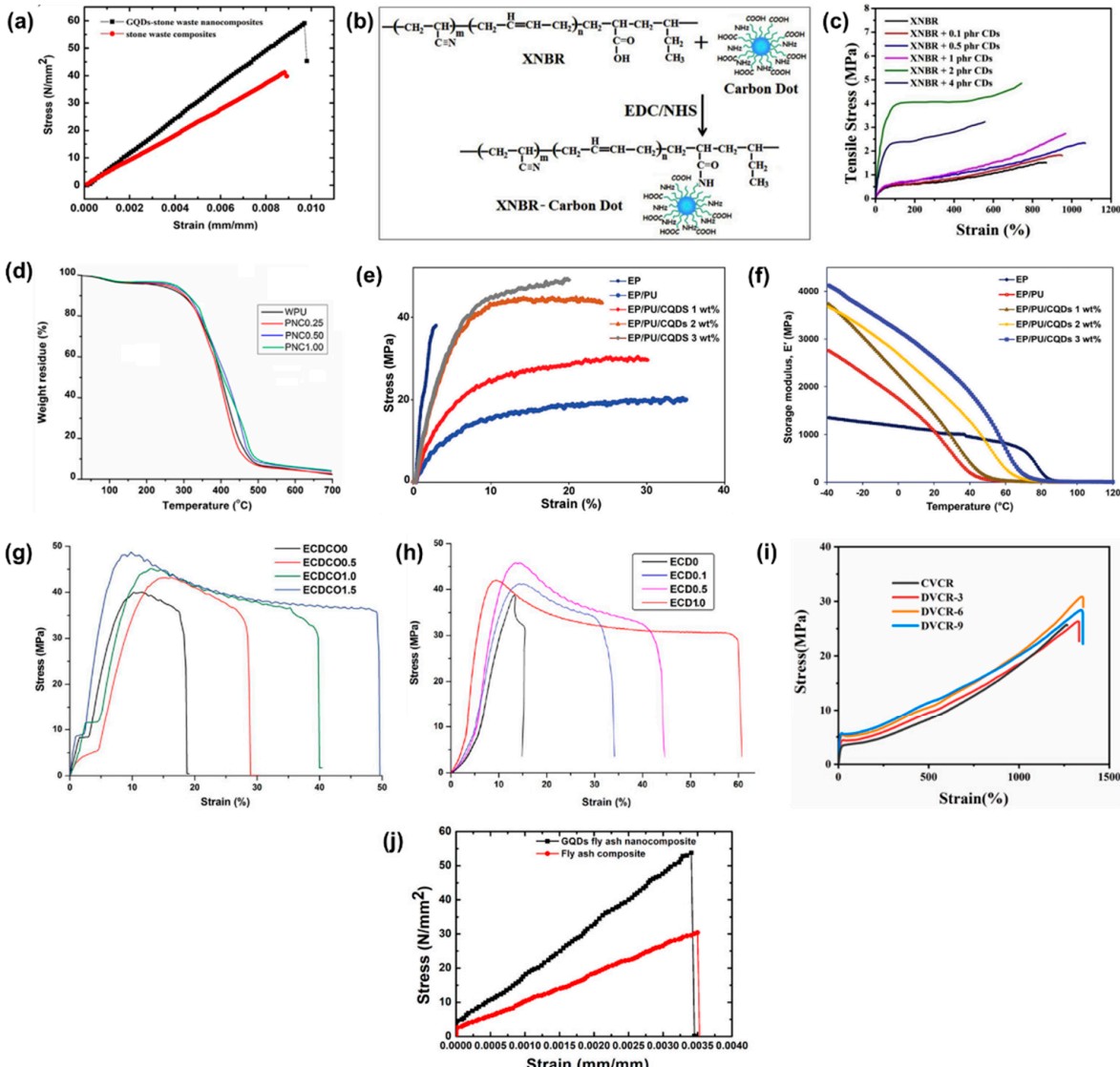

**Figure 7.** (**a**) Stress–strain curves of stone waste composite and CQD-reinforced stone waste nanocomposite [90], (**b**) stepwise fabrication of XNBR/CQD nanocomposite [91], (**c**) stress–strain curves of XNBR/CQDs with various loadings of CQDs [91], (**d**) thermal stability of pure WPU and WPU/CQD nanocomposites [92]. Mechanical properties of EP, EP/PU, and EP/PU/CQI (**e**) stress–strain curves and (**f**) storage modulus [88]. Stress–strain curves of epoxy/CQD nanocomposite with different contents of CQDs: (**g**) 0 to 1.5% [93] and (**h**) 0 to 1% [94]. (**i**) Stress–strain curves of pure CR and CR/CQDs [95], (**j**) stress–strain curves of fly ash/epoxy and fly ash/epoxy/CQDs nanocomposite [96].

The effect of CQDs at low concentrations on reinforcing carboxylated acrylonitrile butadiene (XNBR) was studied [91]. The coupling chemicals 1-(3-dimethylaminopropyl)-3-ethylcarbodiimide hydrochloride (EDC.HCl) and N-hydroxysuccinimide (NHS) were used to covalently conjugate the CQDs to XNBR (Figure 7b). The incorporation of CQDs into XNBR significantly improved its tensile stress–strain characteristics, as illustrated in Figure 7c. Compared to pure XNBR, the optimal tensile stress of the nanocomposite was 215% improved with the addition of 2 phr CQDs. With the addition of CQDs to XNBR, there was a simultaneous drop in the tan δ peak. Additionally, compared to pure XNBR, a specimen having 2 phr of CQDs had a 161% higher storage modulus ($G'$). A specimen including 4 phr of CQDs had an initial degradation temperature value that was 40 °C greater compared to that of pure XNBR. On the contrary, XNBR having 1 phr of CQDs had a maximum degradation temperature that was 11 °C greater than that of pure XNBR [91].

In situ-synthesized waterborne polyurethane (WPU)/CQD nanocomposites were fabricated as potential surface coatings [92]. As CQDs were added, the tensile strength, elongation at break, and hardness of the thermosetting WPU nanocomposites were enhanced by 89%, 42%, and 200%, respectively. In addition, among the thermal properties, the thermal stability of CQD-reinforced WPU nanocomposite increased from 250 to 280 °C, as shown in Figure 7d. As a result, the nanocomposites exhibit high potential as transparent surface-coating materials with low volatile organic compound content [92].

The characteristics of smart hybrid coatings based on epoxy/PU reinforced with CQDs were analyzed [97]. They were prepared by combining epoxy resin with a PU pre-polymer to fabricate flexible and transparent polymer coatings. Tensile strength, flexibility, and viscoelastic characteristics, including the storage modulus, were improved by 150%, 35%, and 209%, respectively, with the addition of the optimal CQD content, as shown in Figure 7e,f. According to the results of a salt spray test, coatings with higher CQD concentrations exhibit better anti-corrosive properties [97].

A thermostable hyperbranched epoxy nanocomposite was prepared with the inclusion of a carbon-dot-reduced $Cu_2O$ nanohybrid as the nano-reinforcement [94]. By reducing the cupric acetate solution with carbon dots at 70 °C for 6 h, $Cu_2O$ nanohybrid particles were synthesized. The nanocomposite of nanohybrids and hyperbranched epoxy was fabricated to analyze the mechanical and thermal performance. The development of a nanocomposite with the inclusion of 1.5 wt.% nanohybrid resulted in a suitable enhancement in the performance, including 20% higher tensile strength (Figure 7g), 2.5 times higher elongation at break, 3.5 times higher toughness, and a temperature stability increase of 23 °C as compared to pure epoxy [94].

The inclusion of CQDs led to the fabrication of a luminous transparent hyperbranched epoxy nanocomposite with excellent toughness and elasticity [98]. An in situ solution method performed by curing with poly(amido-amine) at 100 °C was used to fabricate nanocomposites of hyperbranched epoxy with CQDs at various CQD contents. With only 0.5 wt.% CQDs, the toughness of pure epoxy was increased by 750%. The elongation at break was significantly enhanced from 15 to 45%, and the tensile strength was enhanced from 38 to 46 MPa, as shown in Figure 7h [98].

Nitrogen-doped carbon nanodots (N-CQDs) were used as a crosslinker and reinforcing nanofiller in chloroprene rubber (CR) to fabricate a mechanically durable CR/N-CQDs nanocomposite [99]. The N-CQDs were synthesized using a large-scale, environmentally friendly approach and high-temperature carbonization that can crosslink CR by nucleophilic substitution. Additionally, the covalent crosslinks enable interfacial bonding between N-CQDs and CR, allowing for superior rubber matrix reinforcement. N-CQDs crosslinked with CR exhibit exceptional mechanical characteristics, with a tensile stress of up to 30 MPa and an elongation at break of up to 1354%, as shown in Figure 7i [99].

Paddy straw agricultural waste was employed to synthesize crystalline CQDs by a hydrothermal process, and then CQD-reinforced fly-ash polymer nanocomposites were fabricated [100]. The fly-ash/epoxy/CQD nanocomposite was fabricated using a compression molding technique. In contrast to the 35 MPa flexural strength of the pure fly-ash polymer

hybrid composite, the CQD-embedded fly-ash-based nanocomposites had a high flexural strength of 60 MPa (Figure 7j). The increased interfacial and chemical bonding of crystalline CQDs with fly ash and epoxy is the cause of the flexural strength of the agricultural-waste-derived CQDs incorporated into the fly-ash polymer nanocomposite [100].

In another study, nanocomposites of CQDs and bioresorbable PLA were fabricated to analyze their mechanical performance [101]. The PLA-CQD nanocomposite was 3D-printed to achieve better structural integrity and mechanical and biological characteristics for cardiovascular structures. Compared to pure PLA, the PLA-CQD composites showed increases in tensile strength and compressive strength of 24% and 66%, respectively. According to the findings, adding CQDs considerably improves properties including tensile and compressive strength when compared to pure PLA [101].

Electrospun carbon nanofibers (ECNFs) were formed using polyacrylonitrile (PAN) and coal-based carbon quantum dots (C-CQDs) [93]. Adding C-CQDs to a spinning solution may greatly increase the tensile strength, Young's modulus, and flexibility of the ECNFs. When compared to pure polyacrylonitrile-derived ECNFs, the CGQD-added composite has a Young's modulus that is more than 7-fold greater [93]. The inclusion of C-CQDs promotes the formation of smaller-sized crystalline domains, which drastically improves the toughness of composite PAN fibers.

High-performance TPU/CQD nanocomposites with strong luminescence were prepared using in situ polymerization [95]. The thermal characteristics of TPU/CQDs were examined. The structure–property correlations of the composites were established through their rheological characteristics. The nanocomposites' viscoelasticity was improved with the incorporation of the CQDs. The tensile strength of the TPU/CQD nanocomposite was enhanced by 57% with the inclusion of 1.0 wt.% CQDs as compared to pure TPU [95].

Microwave-assisted N-CQDs synthesized from alginate serve as a drug delivery system and a toughening agent for hydrogels [96]. N-CQDs can function as a toughening agent and as a viscosity modifier for poly(acrylic acid-co methacrylamide) copolymer hydrogels. The hybrid hydrogels demonstrated a low permanent set and were mechanically resilient with 1200% higher elongation at break. Using vacuum estimation, the crosslinked structure was analyzed. The results show that as the N-CQD concentration rises, the network gradually becomes denser [96].

Recent works related to carbon-quantum-dot-based polymer nanocomposites are summarized in Table 5.

**Table 5.** Summary of recent works related to carbon quantum dots-based polymer nanocomposites.

| Sr. | Polymer | Percentage | Technique | Improvement | Reference |
|-----|---------|-----------|-----------|-------------|-----------|
| | | | **Carbon-Quantum-Dot-Based Nanocomposites** | | |
| 1 | Polyester | 5 wt.% | Mixing | Mechanical properties: 62% | [87] |
| 2 | Epoxy/PU | 1–3 wt.% | Mixing | Mechanical properties: 32% | [88] |
| 3 | PEO | N/A | Mixing | Mechanical properties: 157% | [89] |
| 4 | Stone waste | N/A | Mixing | Mechanical properties: 58% | [90] |
| 5 | XNBR | 2 phr | Stirring | Thermal properties: 40 °C Mechanical properties: 161% | [91] |
| 6 | WPU | 1 wt.% | Mixing | Thermal properties: 12% Mechanical properties: 89% | [92] |
| 7 | Epoxy/PU | N/A | Mixing | Mechanical properties: 150% | [97] |
| 8 | Epoxy | 1.5 wt.% | Magnetic Stirring | Thermal properties: 23 °C Mechanical properties: 20% | [94] |
| 9 | Epoxy | 0.5 wt.% | Magnetic Stirring | Mechanical properties: 21% | [98] |
| 10 | CR | N/A | Melt-blending | Mechanical properties: 460% | [99] |

**Table 5.** *Cont.*

| Sr. | Polymer | Percentage | Technique | Improvement | Reference |
|---|---|---|---|---|---|
| | | | Carbon-Quantum-Dot-Based Nanocomposites | | |
| 11 | Epoxy | N/A | Mixing | Mechanical properties: 120% | [100] |
| 12 | PLA | N/A | Mixing | Mechanical properties: 66% | [101] |
| 13 | PAN | 1 wt.% | Stirring | Mechanical properties: 7 folds | [93] |
| 14 | TPU | 1 wt.% | Stirring | Mechanical properties: 57% | [95] |
| 15 | Poly (acrylic acid-co-meth acrylamide) | 2 wt.% | Stirring | N/A | [96] |

## 7. Fullerene-Based Nanocomposites

Fullerene is a versatile carbon nanomaterial that has recently been used in polymer matrices as a reinforcement material. Fullerene is an excellent nanomaterial that has synergistic impacts on improving physical properties. Thus, its design, dispersion, and properties, including thermal stability and mechanical consistency, have been examined. For instance, the mechanical properties of virgin and modified fullerene (oxidized fullerene) with an epoxy matrix were analyzed [102]. Through ultrasonic mixing, fullerene was dispersed in epoxy resin. According to the studies, adding pure fullerene to an epoxy matrix increases the tensile strength and Young's modulus by 14% and 17%, respectively, compared to pure epoxy. Adding oxidized fullerene increases the tensile strength and Young's modulus by 23% and 46%, respectively, as shown in Figure 8a. Nevertheless, because of the addition of oxidized fullerene, the amount of elongation and toughness decreased; moreover, the fracture toughness and strain energy release rate increased by 102% and 86%, respectively [102].

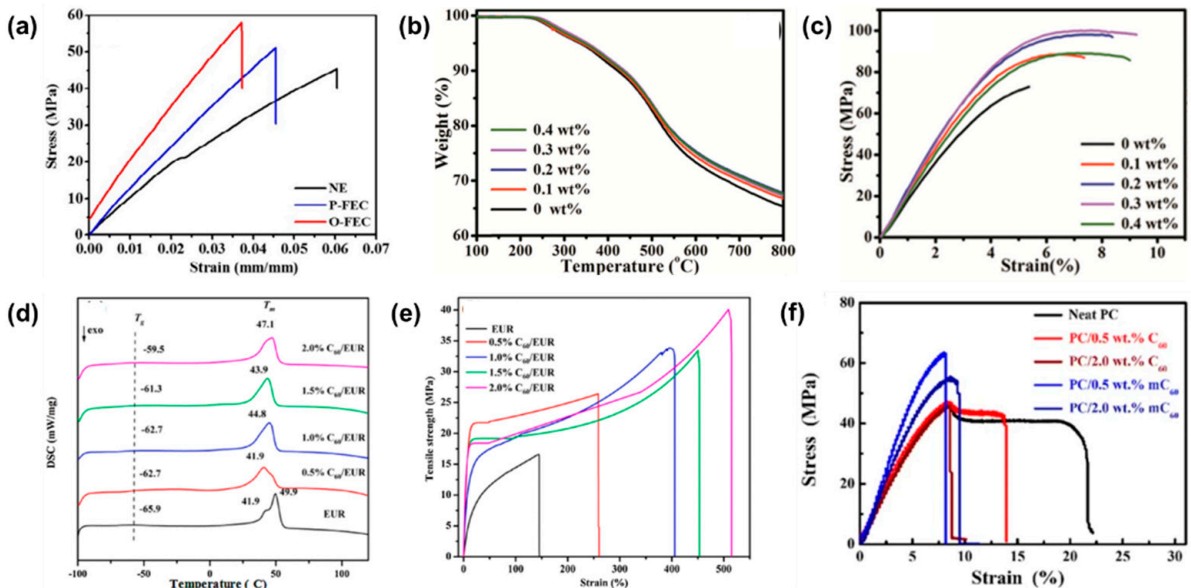

**Figure 8.** (**a**) Stress–strain curves of pure epoxy and $C_{60}$/epoxy composite [102], (**b**) thermal resistance of $C_{60}$/fPA [103], (**c**) stress–strain curves of $C_{60}$/fPA [103] (**d**) thermal properties of EUR and $EURI_0$, [104] (**e**) tensile strength of pure EUR and EUR/$C_{60}$ [104], (**f**) stress–strain curves of neat PC and PC/$C_{60}$ nanocomposites [105].

Reversibly crosslinked composites that were both mechanically robust and thermally stable were fabricated using fullerene ($C_{60}$) and aromatic polyamide (fPA) containing pendant furan groups [103]. At 0.3 wt.% $C_{60}$, the Diels–Alder (D-A) reaction between furan

groups and $C_{60}$ enabled the crosslinking of fPA at an optimal temperature, which improved the thermal and mechanical properties by 12 °C and 43%, respectively (Figure 8b,c). Crosslinked $C_{60}$/fPA was able to be dissociated and recycled at high temperatures thanks to the D-A reaction. The suitable recyclability of $C_{60}$/fPA composites was confirmed by a tensile test, which showed that the mechanical characteristics of regenerated composites were comparable to those of the initial specimens. Additionally, it was shown that $C_{60}$/fPA composites have better high-temperature shape memory than pure fPA [103].

A Eucommia ulmoides rubber (EUR) matrix was treated with ultrasonic energy while various quantities of fullerene $C_{60}$ were added to create nanocomposite films [104]. The thermal and mechanical characteristics of these nanocomposites were also thoroughly examined. Adding $C_{60}$ improved the melting temperature by 12.5% and mechanical performance by 67% as compared to pure EUR, as illustrated in Figure 8d,e [104].

A nanocomposite of fullerene and polycarbonate (PC) was fabricated using the solution-casting method, where $C_{60}$ and modified $C_{60}$ work as the nucleating agents of PC [105]. The PC/modified $C_{60}$ showed homogeneous matrix dispersion and promoted the production of microcrystalline structures. The mechanical characteristics of the PC/modified $C_{60}$ nanocomposites were found to be significantly improved by the suitable degree of crystallinity (Figure 8f). When compared to pure PC films, the PC/0.50 wt.%-modified $C_{60}$ nanocomposites often show up to a 33% improvement in tensile strength and 45% enhancement in Young's modulus [105].

A nanocomposite of fullerene/sulfonated polyetherimide (SPEI) was fabricated to enhance the electromechanical properties of the actuator [106]. The bending distortion of the fullerene/SPEI actuators was improved three-fold by the addition of 0.5 wt.% fullerene reinforcement in the SPEI matrix. The as-prepared actuators had an enhancement in mechanical strength of 114% with the introduction of 0.5 wt.% fullerenes compared to pure SPEI [106].

A fullerene-reinforced PMMA nanocomposite was fabricated to study the mechanical properties of fullerene/PMMA [107]. Molecular dynamics (MD) models were used to determine Young's moduli of nanocomposites containing various fullerene weight fractions. A micromechanics model of a composite with multiple additions of nanoparticles was proposed using the Mori–Tanaka and Eshelby models. When the micromechanics model's numerical findings were compared to those computed using MD simulations, excellent performance was found [107]. The Young's modulus of PMMA/fullerene nanocomposites improved by 25% with the inclusion of 4 wt.% fullerene in the PMMA matrix [107,108].

Films of $C_{60}$/PVDF polymer nanocomposites were fabricated by the solution-casting method [109]. Pristine PVDF and $C_{60}$-PVDF polymer nanocomposite films' optical, microstructural, thermal, and mechanical characteristics were investigated. In PNC films containing 0.5 wt.% $C_{60}$, the degradation temperature increased from 444 °C in pure PVDF to 507 °C, which is attributed to crosslinking between $C_{60}$ and the polymer. The Young's modulus and thermal conductivity of PNC films fluctuate nonlinearly when changing the $C_{60}$ concentration. Crosslinked PVDF particles and agglomerated $C_{60}$ particles with a diameter of around 550 nm are seen in microscopic images [109]. Recent works related to fullerene-based polymer nanocomposites are summarized in Table 6.

**Table 6.** Summary of recent works related to fullerene-based polymer nanocomposites.

| Sr. | Polymer | Percentage | Technique | Improvement | Reference |
|---|---|---|---|---|---|
| **Fullerene Based Nanocomposites** | | | | | |
| 1 | Epoxy | 0.5 wt.% | Mechanical stirring | Mechanical properties: 23% | [102] |
| 2 | PA | 0.3 wt.% | Sonication | Thermal properties: 12 °C <br> Mechanical properties: 43% | [103] |
| 3 | EUR | 2 wt.% | Stirring | Thermal properties: 12.5% <br> Mechanical properties: 67% | [104] |

**Table 6.** *Cont.*

| Sr. | Polymer | Percentage | Technique | Improvement | Reference |
|-----|---------|-----------|-----------|-------------|-----------|
| | | | **Fullerene Based Nanocomposites** | | |
| 4 | PC | 1 wt.% | Stirring | Mechanical properties: 33% | [105] |
| 5 | SPEI | 0.5 wt.% | Magnetic stirring | Mechanical properties: 114% | [106] |
| 6 | PMMA | N/A | Mixing | Mechanical properties: 25% | [107] |
| 7 | PVDF | 0.5 wt.% | Electrical stirring | Thermal properties: 14% | [109] |

## 8. Metal–Organic Frameworks Based Nanocomposite

MOFs are a new family of porous inorganic–organic high-performance nanomaterials that have received significant consideration. MOFs are superior materials in several essential uses because of their higher specific surface area, higher porosity, homogeneous structural nanoscale cavities, and temperature stability [110]. For instance, a laser-sintered PLLA scaffold was filled with zeolitic imidazolate framework-8 (ZIF-8), and one common MOF was studied [111]. The findings demonstrated that ZIF-8's intrinsic hydrophobicity and many organic ligands led to favorable interfacial interaction with PLLA. In addition, it improved the crystallinity of PLLA and acted as an effective nucleating site. As a result, there was an improvement in the tensile and compressive strength of 37% and 86%, respectively (Figure 9a,b) [111].

A paper-based composite having higher mechanical performance was formed using cotton-pulp-based paper with carboxylated cellulose nanofiber (CNF) through the reaction [112]. As a result, zeolitic imidazolate framework-67 (ZIF-67) was formed on the surface of the obtained cellulosic nanomaterial. The mechanical properties of the resultant nanocomposite were analyzed. The tensile strength of the composite with the incorporation of 2.5 wt.% CNF was enhanced by 1.3 times and 11 times under dry and wet conditions, respectively (Figure 9c) [112].

Nanosized UiO-66 and UiO-66-$NH_2$ (type of MOFs) were effectively incorporated into epoxy using a solution-casting technique to fabricate MOF/epoxy composites [113]. The mechanical properties, including the elongation at break and tensile strength, of epoxy/MOF were enhanced by 37% and 13%, respectively (Figure 9d). The findings demonstrated that MOFs, namely, the functioned amino groups, are crucial for toughening epoxy-based nanocomposites [113].

A green synthesis technique was employed for the in situ growth of MOF nanocrystals on wood substrates [114]. Sodium hydroxide treatment forms the nucleation sites for diverse kinds of MOFs, which are highly suitable for many wood species. When compared to natural wood, the resultant MOF/wood composite has 130 times higher specific surface area and shows hierarchical porosity. The evaluation of the $CO_2$ adsorption capacity shows the effective use of MOF contents along with similar adsorption capabilities to pure MOFs. Superior mechanical properties were revealed, with a compression strength that was 47% higher compared to the pure form, as shown in Figure 9e,f. The manufacturing of multifunctional MOF/wood-derived composites with prospective environmental and energy applications is made possible by the functionalization strategy, which provides a robust, environmentally friendly, and cost-effective substrate [114].

Metal–organic frameworks (HKUST-1) were specifically chosen as multifunctional nanofillers to modify the PEO-based solid polymer electrolyte (SPE) [115]. The ionic conductivity of the PEO/lithium-ion transfer (LiTFSI-10)/HKUST-1 electrolyte (PL10HM) was enhanced by up to $2.4 \times 10^{-3}$ S·cm$^{-1}$ at 80 °C. In comparison to the PEO/LiTFSI electrolyte (PL), the electrochemical stability and LiTFSI-10 number were significantly improved using HKUST-1 MOFs. Systematic characterization of the thermal resistance and mechanical characteristics showed that they were improved by 180% and 426%, respectively (Figure 9g,h) [115].

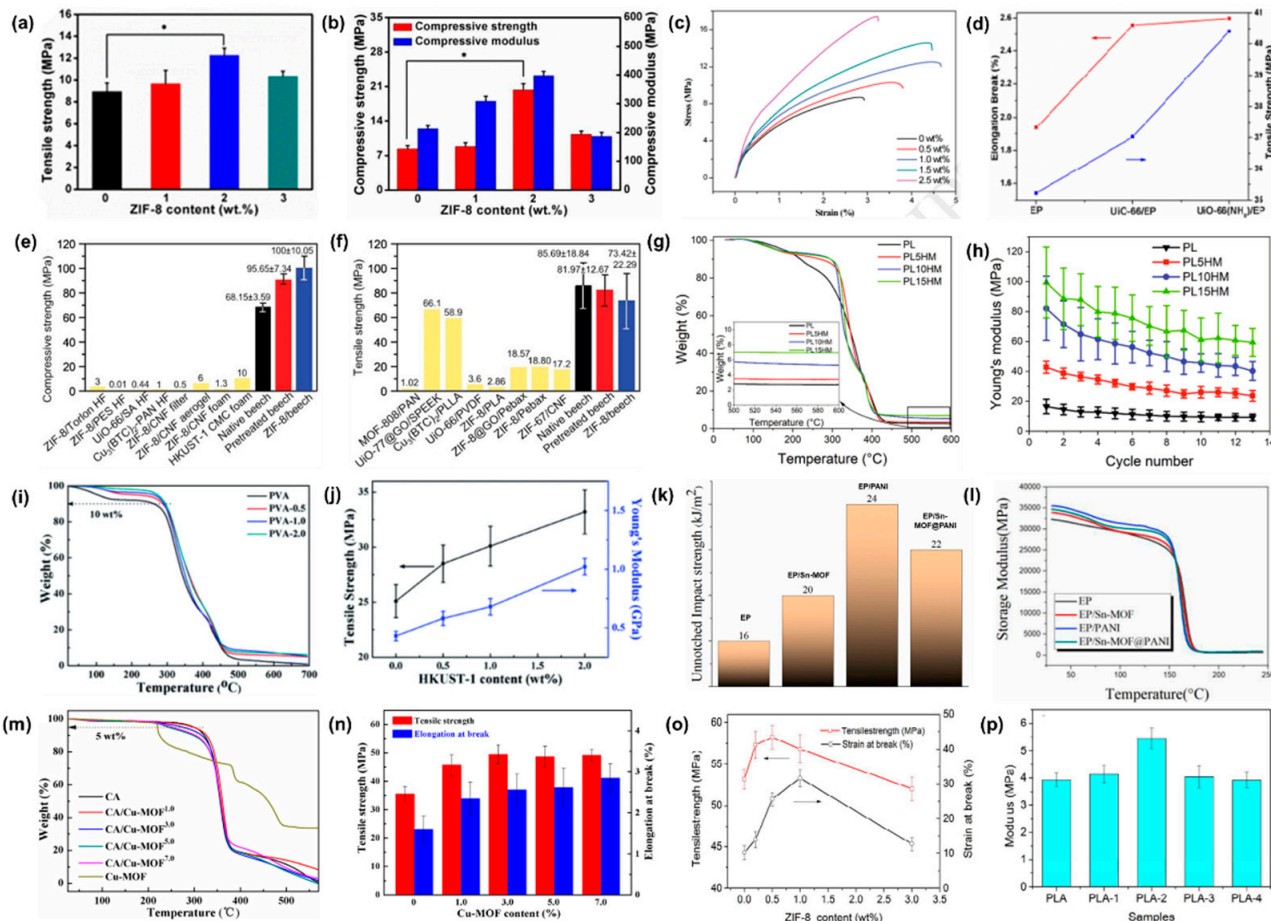

**Figure 9.** Mechanical properties of MOF-reinforced nanocomposites: (**a**) tensile strength of PLLA/ZIF-8 [111], (**b**) compressive strength and modulus of PLLA/ZIF-8 ("*" represents the highest and lowest values [111], (**c**) stress–strain curves of ZIF-67-reinforced nanocomposites [112], (**d**) tensile strength and elongation of EP/MOF [113], (**e**) compressive strength of wood/MOF nanocomposites [114], (**f**) tensile strength of wood/MOF nanocomposites [114]. (**g**) The thermal properties of PL/MOF [115], (**h**) Young's modulus of PL/MOF with respect to cyclic load [115], (**i**) thermal resistance of pure PVA and PVA/MOF nanocomposites [116], (**j**) tensile strength and Young's modulus of pure PVA and PVA/MOF nanocomposites [116], (**k**) impact strength of MOF-reinforced nanocomposites [117], (**l**) storage modulus of MOF-reinforced nanocomposites [117], (**m**) thermal resistance of CA and CA/MOF with different loadings of MOF [118] (**n**) tensile strength and elongation at break of CA and CA/MOF with different loadings of MOF [118], (**o**) tensile strength and strain at break of PLA/ZIF-8 [119], and (**p**) modulus of PLA/ZIF-8 [119].

A facile solution-casting technique was employed to prepare nanocomposites with PVA as the matrix and HKUST-1 as the nano-reinforcement [116]. Because of the uniform dispersion of HKUST-1 and the strong interfacial bonding between PVA and HKUST-1, the inclusion of HKUST-1 significantly improved the thermal and mechanical characteristics of PVA/HKUST-1. The thermal resistance of PVA/HKUST-1 was improved by 33 °C compared to pure PVA with the addition of 2 wt.% HKUST-1 (Figure 9i). In addition, the nanocomposite's Young's modulus and tensile strength were improved by 137% and 32%, respectively, compared to neat PVA (Figure 9j) [116].

Three-dimensional nano-structural MOFs@polyaniline (Sn-MOF@PANI) with epoxy resin were fabricated [117]. The peak heat release rate and total heat release of the epoxy/Sn-MOF@PANI nanocomposite were decreased by 42% and 32%, respectively, when compared to pure epoxy. Throughout the combustion, the total smoke production and total gaseous products of epoxy/Sn-MOF@PANI both decreased. The fire retardancy of the

epoxy composite was also increased. The gaseous phases and the condensed phase led to the suggestion of a potential flame-hindering process. Additionally, the inclusion of Sn-MOF@PANI enhanced the mechanical properties of epoxy, including its impact strength and storage modulus, as shown in Figure 9k,l [117].

Using a solution-casting technique, flower-like MOFs (Cu-MOF) were prepared and added to cellulose acetate (CA) to fabricate functional nanocomposites [118]. The suitable changes in the thermal and mechanical characteristics of CA films with the inclusion of Cu-MOF were analyzed. The results show that the Cu-MOF was compatible with CA and formed homogeneous and compact nanocomposite films. In comparison to the pristine CA film, the as-prepared nanocomposite films exhibited higher thermal ($T_m$) and mechanical characteristics, which increased by 5% and 43%, respectively (Figure 9m,n). The findings suggest that the transparent packaging uses of the CA/Cu-MOF with clearly enhanced physical and functional capabilities offer substantial promise [118].

In another study, PLA/ZIF-8 nanocomposites were fabricated via solution-blending and film-casting techniques [119]. ZIF-8 nanoparticles were added to the polymer, which enhanced the strain at break by 200% with the addition of 1 wt.% ZIF-8, and the modulus was increased by 42% with 2 wt.% ZIF-8 (Figure 9o,p). Even when the ZIF-8 nanoparticle concentration increased to 3 wt.%, the produced PLA/ZIF-8 films maintained excellent visibility [119].

The incorporation of MOFs as reinforcement in SBR via a solvothermal reaction between 2-amino terephthalic acid and aluminum chloride hexahydrate was analyzed [120]. A comparative analysis of the characteristics of a uniformly dispersed and thermally resistant nano-MOF composite (SBR-MOF) was conducted with reference to the SBR-nano-alumina composite (SBR-nAl). SBR-MOF offered higher mechanical durability than SBR-nAl. SBR-MOF demonstrated 50% greater elongation at break than SBR-nAl at 10 phr and a 130% increase in tensile strength over the pure SBR composite. SBR-MOF has better thermal and dynamic mechanical characteristics than the SBR-nAl composite. The porous MOFs encouraged greater polymer chain entangling at the interface [120].

Stretchable sensors made of ionogel are frequently used in soft robotics and wearable healthcare applications [121]. The metal interaction and porosity of MOFs create physically bonded ionogels that are made of polymers that align with the MOF metal spots. The material's covalent bonding was turned into reversible bonding that considerably improved the MOF-ionogels' mechanical characteristics. The resulting ionogels had a Young's modulus and toughness of 58 MPa and 25 MJ·m$^{-3}$, respectively, and could withstand an 11,000% stretch. Additionally, the fracture energy was higher than most observed ionogels, reaching 125 kJ·m$^{-2}$ [121].

Ti-metal–organic framework nanofiber (MOFNF) was fabricated by electrospinning to study the characteristics of MOF/nanofiber nanocomposites [122]. A fractional factorial design was employed to analyze the thermodynamic and mechanical properties of the as-prepared nanocomposites. MOFNF was employed as a dentistry nano-coating with a surface area of around 3204 m$^2$/g, thermal resistance up to 370 °C, and 560 Pa compressive strength. Unlike conventional nano-coatings, appropriate mechanical and physicochemical characteristics were achieved [122].

Recent works related to metal–organic framework-based polymer nanocomposites are summarized in Table 7.

**Table 7.** Summary of recent works related to metal–organic framework-based polymer nanocomposites.

| Sr. | Polymer | Percentage | Technique | Improvement | Reference |
|---|---|---|---|---|---|
| **Metal–Organic Framework-Based Nanocomposites** | | | | | |
| 1 | PLLA | 2 wt.% | Mixing | Mechanical properties: 37% | [111] |
| 2 | Epoxy | 2.5 wt.% | Mixing | Mechanical properties: 37% | [113] |

**Table 7.** *Cont.*

| Sr. | Polymer | Percentage | Technique | Improvement | Reference |
|---|---|---|---|---|---|
| | | | **Metal–Organic Framework-Based Nanocomposites** | | |
| 3 | PEO | 15 wt.% | Mixing | Thermal properties: 180% Mechanical properties: 426% | [115] |
| 4 | PVA | 2 wt.% | Stirring | Thermal properties: 33 °C Mechanical properties: 32% | [116] |
| 5 | Epoxy | 2 wt.% | Sonication | Mechanical properties: 42% | [117] |
| 6 | CA | 7 wt.% | Stirring | Thermal properties: 5% Mechanical properties: 43% | [118] |
| 7 | PLA | 1 wt.% | Stirring | Mechanical properties: 42% | [119] |
| 8 | SBR | 10 phr | Mixing | Mechanical properties: 130% | [120] |

## 9. Summary and Outlook

Nanocomposites fabricated using different derivatives of nano-carbon as nano-reinforcements in polymer matrices have demonstrated superior mechanical and thermal characteristics. The characteristics of nano-carbon reinforcements that enhance the properties of nanocomposites include (i) a high aspect ratio, as well as strong interfacial interactions with the matrix, (ii) homogeneous dispersion of nanofillers in the matrix, (iii) higher bonding strength between nanofillers and the polymer, (iv) crosslinking behavior of nanofillers with matrices, (v) greater mechanical interlocking of nanofillers and matrices, (vi) strong intermolecular interactions of these nanofillers with polymer matrices, suppressing free vibrations and hence reducing heat conduction. On the basis of these characteristics, the mechanical and thermal properties of nano-carbon-reinforced polymer nanocomposites can be significantly enhanced, as highlighted in this review. The performance of nano-carbon-reinforced polymer nanocomposites in different domains of use is summarized in tables.

The mechanical properties of nanocomposites with the inclusion of nano-carbon, including graphene, MXenes, CNTs, CB, CQDs, fullerene, and MOFs, were enhanced by 60%, 80%, 100%, 50%, 200%, 100%, and 137%, respectively, with the optimal content of nano-carbon. The maximum increase in the mechanical properties of polymer nanocomposites was 200% with the optimal inclusion of CQDs. CQDs offer higher mechanical properties with polymer matrices; however, their production yield on a gram scale is challenging. The synthesis precursors of CQDs are abundant and cost-effective, while the synthesis method is not economically beneficial. Furthermore, CB is an abundant and low-cost nano-carbon material that is synthesized on a gram scale; this scalability of CB is more economically beneficial.

The thermal stability of polymer nanocomposites with the inclusion of graphene, MXenes, CNTs, CB, CQDs, fullerene, and MOFs increased by 60%, 40%, 20%, 30%, 5%, 14%, and 33% respectively, with the optimal incorporation of nano-carbons. The thermal stability of polymer nanocomposites was increased by 60% with the optimal addition of graphene to polymer matrices. Graphene offers higher thermal stability with polymer matrices; however, different toxic and expensive chemicals are used for their synthesis. These chemicals make graphene toxic and expensive, which is not beneficial from environmental and economic perspectives. Moreover, CB is a non-toxic and low-cost abundant nanoparticle that considerably improves the thermal stability of polymers. The cost-effectiveness, scalable production yield, and non-toxicity of CB are highly suitable candidates for the modification of polymers.

## 10. Future Perspectives

While some carbon nanomaterials offer higher mechanical and thermal properties to polymer nanocomposites, the scalable synthesis methods of some nano-carbons are still

challenging. Due to their high production costs, these carbon nanomaterials cannot be applied on a commercial scale. Additionally, the surface functionalization of nano-carbon is significant for thermal properties, which is a research gap that can be addressed in future research. Furthermore, limited studies have focused on the recyclability of nano-carbon-reinforced composites; therefore, it will be necessary to increase the development of recycled nanocomposites. A promising method would be the use of waste thermoset-polymer-based nano-carbon-reinforced composites to develop high-added-value carbon nanomaterials.

**Author Contributions:** Conceptualization: Z.L. and M.A.; methodology: M.A.; software: Z.L.; validation: E.-J.L. and Z.Z.; formal analysis: Z.Z.; investigation: M.A.; resources: K.H.L.; data curation: Z.L.; writing—original draft preparation: Z.L.; writing—review and editing: M.A.; visualization: Z.L.; supervision: Z.Z. and K.H.L.; project administration: K.H.L.; funding acquisition: K.H.L. All authors have read and agreed to the published version of the manuscript.

**Funding:** This work was supported by Korea Environment Industry and Technology Institute (KEITI) through Environmental R&D Project on the Disaster Prevention of Environmental Facilities Project, funded by Korea Ministry of Environment (MOE) (2020002870004).

**Data Availability Statement:** Not applicable.

**Acknowledgments:** This work is carried out under the umbrella of GCF-63, funded by Higher Education Commission, Pakistan. We highly appreciate the help of Kinza Shahid in graphical abstract design.

**Conflicts of Interest:** The authors declare no conflict of interest.

## List of Abbreviations

CNTs = Carbon nanotubes
CQDs = Carbon quantum dots
MWCNTs = Multi-walled carbon nanotubes
1D = One-dimensional
GO = Graphene oxide
PMMA = Poly methyl methacrylate
PVA = Polyvinyl alcohol
MFC/NFC = Micro- and nano-fibrillated cellulose
CNC = Cellulose nanocrystal
PiP-DOPO = Piperazine DOPO-phosphonamidate
HQ = Hydroquinon
GPOSS = Glycidyl isooctyl polyhedral oligomeric silsesquioxane
PBT = Polybutylene terephthalate
PU = Polyurethane
SMPNCs = Shape memory polymer nanocomposites
Zr-AMP = Zirconium amino-tris-(methylene phosphonate)
PLA = Polylactic acid
PP = Polypropylene
NR = Natural rubber
CNTPN = Carbon nanotube polymer nanocomposites
FCNTPN = Functionalized carbon nanotube polymer nanocomposites
ICNTPN = Intact carbon nanotube polymer nanocomposites

CB = Carbon black
SWCNTs = Single-walled carbon nanotubes
2D = Two-dimensional
0D = Zero-dimensional
rGO = Reduced graphene oxide
MMA = Methyl methacrylate
PVDF = Polyvinylidene fluoride
FTIR = Fourier transform infrared
LOI = Limiting oxygen index
PF = Phenol formaldehyde
PNMTh = Poly N-methylthionine
NaCMC = Sodium carboxymethyl cellulose
CMC = Carboxymethyl cellulose
PET = Polyethylene terephthalate
PCL = Poly caprolactone
FGO = Functionalized graphene oxide
SME = Shape memory effect
TPU = Thermoplastic polyurethane
MNH = MXene nanoscale hydrogel
MNOH = MXene nanocomposite
PPDA = Phenyl phosphonic diamine hexane
MCA = Melamine cyanurate
NBR = Nitrile butadiene rubber
PSS = Polystyrene sulfonate
CPNC = Carbon polymer nanocomposite
PDA = Polydopamine
HDPE = High-density polyethylene
DDA = Dodecyl amine
MCNTs = Melamine carbon nanotubes
SENB = Single-end notch bend

PNCs = Polymer nanocomposites
GSD = Graphic structure design
GIc = Mode I interlaminar fracture toughness
GFRPs = Glass-fiber-reinforced polymers
SBR = Styrene-butadiene rubber
IIR = Butyl rubber
EPDMn = Ethylene propylene diene monomer
CVD = Chemical vapor deposition
IFSS = Interphase shear strength
XNBR = Carboxylated acrylonitrile
butadiene rubber
EDC HCl = Ethyl carbodiimide hydrochloride
ECNFs = Electrospun carbon nanofiber textiles
CGQDs = Coal-based graphene quantum dots
N-CQDs = Nitrogen-doped carbon quantum dots
EUR = Eucommia ulmoides rubber
PEPA = Polyethylene polyamine
FRPNs = Fullerene-reinforced polymer
nanocomposites
PLLA = Poly-L-lactic acid
SBR-nAl = Styrene-butadiene rubber
nano-alumina
MOFNF = Metal–organic framework nanofibers

PCNT = Polymer carbon nanotube
CFRPs = Carbon-fiber-reinforced polymers
FMLs = Fiber metal laminates
ABS = Acrylonitrile–butadiene–styrene
IPN = Interpenetrating polymer network
PE = Polyethylene
CF = Carbon fiber
ILSS = Interlaminar shear strength
PVC = Polyvinyl chloride
PEO = Polyethylene oxide
NHS = N-hydroxy succinimide
WPU = Waterborne polyurethane
CR = Chloroprene rubber
fPA = Aromatic polyamide
D-A = Diels–Alder
SPEI = Sulfonated polyetherimide
ED = Epoxy diane
PC = Polycarbonate
MD = Molecular dynamics
ZIF-8 = Zeolitic imidazolate framework-8
CNF = Carboxylated cellulose nanofibers
SPE = Solid polymer electrolytes
CA = Cellulose acetate

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
