# Peer review of "Thermal and Mechanical Properties of Nano-Carbon-Reinforced Polymeric Nanocomposites: A Review"

_jcs, doi:10.3390/jcs7100441_

Round 1

Reviewer 1 Report

This manuscript has the ambition to review the mechanical and thermal properties improvements of three primary types of polymer nanocomposites, based on:  two-dimensional (2D) including graphene (rGO) and MXene; one-dimensional (1D) including different types of carbon nanotubes (CNTs); and zero-dimensional (0D) including carbon quantum dots (CQDs), fullerene, and carbon black.

The results from 119 publications are summarized, in respect to the percentage of improvement of mechanical and thermal properties at the optimal filler contents, of composites based on different matrix polymers and carbon fillers, produced by different processing methods.

Six parameters of nano-carbon reinforcements to enhance nanocomposites are pointed in the Introduction (Page 2) and in the Summary (Page 24), as folows:  i) high aspect ratio as well as large interfacial interactions with the polymer chains, ii)   homogenous dispersion of nanofillers in the polymer, iii) higher bonding strength between nanofillers and polymer, iv) crosslinking behavior of nanofillers with matrices,  v)  greater mechanical interlocking of nanofillers and matrices,  vi) strong intermolecular interaction of these nanofillers with polymer matrices suppress the free vibrations and hence reduced heat conduction. 

Comments and suggestions:

1) This review manuscript pointed only informatively the effects of the upper 6 parameters on the properties improvement, but a serious analysis for their role and effectiveness is missing.

2) It is not clear what are the differences between three of the parameters, described above as: (i) large interfacial interactions with the polymer chains, (iii) higher bonding strength between nanofillers and polymer, and (v)  greater mechanical interlocking of nanofillers and matrices, as in fact they present one and the same effect of the interactions of carbon nanoparticles with the polymer matrices.

3)  Moreover, the effects of the 2D, 1D and 0D nanofillers on the mechanisms of improvement of mechanical and thermal properties is insufficiently described.

4) The analysis of the structure-properties relationships of the referred three primary types of the polymer nanocomposites is missing.

5) In the Summary, authors try to categorize the nanocarbon composites with optimum filler content of graphene, MXene, CNTs, CB, CQDs, fullerene, and MOFs inclusion, by the percentage of enhancement of mechanical and thermal properties. However, this categorization is made only by the type of the nanofiller and the roles of the matrix polymer, the filler-polymer interactions and etc., are not taken into account.  These conclusions need of improvement.

Some English errors are found, so the text is needed of check and corrections.

Reviewer 2 Report

(1) There is a significant amount of whitespace between Figure 1 and the preceding text. The authors are advised to review the formatting and layout to minimize such occurrences in the manuscript.

(2) The arrangement of images within Figure 2 appears somewhat disorganized. The authors are encouraged to make the necessary revisions to enhance the clarity of the figure and to conduct a self-review to identify and rectify similar issues throughout the manuscript.

(3) Could the authors contemplate summarizing and presenting the content of each subsection in the form of tables? This approach could contribute to improving the overall clarity and comprehensibility of the article. The content summarization in Table 1 seems overly condensed and succinct.

(4) Figure 3(g) displays a watermark on the image. The authors are requested to examine this issue and undertake the appropriate corrective measures.

(5) It is suggested that the authors incorporate a higher proportion of recent literature in the references section. This adjustment would not only underscore the timeliness of the article but also facilitate readers in anticipating future directions of research.

Reviewer 3 Report

Nowadays, different forms of nano-carbon materials are widely used to improve thermal and mechanical properties of polymers. Usually, small amount of such material is added to a polymer to form a composite. An attempt to review the recent trends of production and characterization of such a composites is presented in the manuscript. Research in this field is developing fast, and such a composites are of importance for industrial application. Therefore, this review is timely and can be published, in principle. However, a number of drawbacks should be corrected.

First, there is a lack of systematic approach in presenting the results taken from recent publications. This leads to weird statements in the “Summary and outlook” section (please also correct “Summery” in the title of this section). An example of such a statement is “The mechanical properties of nano-carbon including graphene, MXene, CNTs, CB, CQDs, fullerene, and MOFs inclusion nanacompsoites enhanced by 60%, 80%, 100%, 50%, 200%, 100%, and 137% respectively, with optimum content of nano-carbon” (lines 952-954). This statement is imprecise and misleading, because there are many different mechanical properties of materials, for example, Young modulus, fracture toughness, tensile strength, density, velocity of sound and so on. Therefore, mechanical properties can not be characterized by a single number. The same holds for thermal properties. In order to avoid this uncertainty, I suggest to introduce a separate section (for example, after “Introduction” section). The overview of mechanical and thermal properties of polymers which can be improved by nano-carbon additive should be presented in this section. This includes definition of each property, the method of its characterization and importance of this property for practical use of the polymer material.  This section will make the rest of the paper easy for understanding.

Second, none of the figures in the manuscript has suitable quality. Figure 1 should be re-shaped to make visible the structure of the substances typically used in polymer/nano-carbon composites (at present they are too small). Figures 2-9 are completely useless, because they can not be understood without referring to the original articles cited in the manuscript. The panels in these figures are too small, the images and text do not have adequate resolution. Figures 2-9 should be replaced by less crowded pictures. I suggest to select one or two most prominent examples of each type of polymer/nano-carbon composites and present detailed pictures for them.

Third, not only advantages of use of nano-carbon materials in composites with polymers, but also its drawbacks should be explained for each type of the additive.

Minor remarks:

1.     Ref.  49 is a purely theoretical work. Please mark it accordingly. Otherwise the readers will be confused because it is sited along with experimental works without any distinction.

2.     Please explain the meaning of “phr” (line 666 and Table 1). This is not a common abbreviation.

3.     English should be carefully corrected throughout the manuscript. There are many grammatical and spelling errors.

Some examples or errors:

Line 307 and further “interphase” should be replaced by “interface”.

Line 982 “value added” should be replaced by “added value”.

Lines 43-45. Unclear sentence: “The second class of carbon-based nanofillers is one- dimensional (1D) reinforced including carbon nanotubes (CNTs) including their different types…” Probably, the first “including” should be deleted.

There are many other errors.

Round 2

Reviewer 1 Report

The revised manuscript could be published.

Reviewer 3 Report

The authors took into account some recommendations of the reviewers. Althouth the manuscript is not perfect, I do not see how it can be improved further.